# Plug, Play, and Fortify: A Low-Cost Module for Robust Multimodal Image Understanding Models

**Siqi Lu,**[*] **Wanying Xu,**[*] **Yongbin Zheng**[†]**, Wenting Luan, Peng Sun, Jianhang Yao**
National University of Defense Technology
{lusiqi9803, wanying_xu, zybnudt}@nudt.edu.cn
{luanwentingnudt, sunpeng, yaojianhang23}@nudt.edu.cn

## Abstract

Missing modalities present a fundamental challenge in multimodal models, often causing catastrophic performance degradation. Our observations suggest that this fragility stems from an imbalanced learning process, where the model develops an implicit preference for certain modalities, leading to under-optimization of others. We propose a simple yet efficient method to address this challenge. The central insight of our work is that the dominance relationship between modalities can be effectively discerned and quantified in the frequency domain. To leverage this principle, we first introduce a **F**requency **R**atio **M**etric (FRM) to quantify modality preference by analyzing features in the frequency domain. Guided by FRM, we propose a **M**ultimodal **W**eight **A**llocation **M**odule, a plug-and-play component that dynamically rebalances the contribution of each branch during training, thereby promoting a more holistic learning paradigm. Extensive experiments demonstrate that MWAM can be seamlessly integrated into diverse architectural backbones, such as those based on CNNs and ViTs. Furthermore, MWAM delivers consistent performance gains across a wide range of tasks and modality combinations. This advancement extends beyond merely optimizing the performance of the base model; it also yields further improvements to state-of-the-art methods designed to address the missing modality problem.

## 1 Introduction

Multimodal vision understanding models leverage complementary data streams like visible, infrared, and depth images to achieve more robust and accurate inferences. Although the efficacy of multimodal techniques is well established across tasks such as segmentation, detection, and classification, a critical limitation of existing work is the underlying assumption of complete modality availability during inference. This assumption often proves unrealistic in practice, as real-world applications are frequently compromised by challenges such as sensor failure, adverse environmental conditions, or privacy constraints, resulting in scenarios with incomplete modalities (Zhang et al. (2024a)).

Addressing the challenge of missing modalities is crucial for enhancing the robustness of multimodal models. To this end, researchers have proposed several key strategies. One prominent approach involves feature imputation, where the missing modality's features are reconstructed from the available ones to restore a complete multimodal representation and thus mitigate performance degradation (Tran et al. (2017); Lin et al. (2023)). Another distinct strategy bypasses reconstruction and instead projects all modalities into a unified, modality-agnostic feature space (Reza et al. (2024); Lee et al. (2023)). By minimizing inter-modality discrepancies, this method leverages consistent cross-modal information to ensure reliable and stable inference, even when some data is absent.

However, our analysis reveals a critical vulnerability in unified models and standard learning methods: their performance is highly sensitive to the identity of the absent modality. In Table 1, the absence of Depth causes the most severe performance drop, even falling below that of a unimodal model trained solely on the remaining data. Conversely, the absence of RGB has the least impact.

---

[*]Equal Contribution. [†]Corresponding Author. Code: https://github.com/a6103121/MWAM

Table 1: Analysis of model performance under various incomplete modality conditions on the CASIA-SURF dataset. Performance is quantified using Accuracy (Acc) and Performance Collapse Rate (PCR). **Multimodal** refers to models trained on complete data but evaluated with missing modalities during inference. **Uni-modal** indicates models trained and evaluated using only a single modality. **FRM** is the Frequency Ratio Metric detailed in Section 4.2.

| Modal | | | Multimodal | | Uni-modal | | RFNet | | FRM ($\uparrow$) |
|---|---|---|---|---|---|---|---|---|---|
| Rgb | Depth | Ir | Acc($\uparrow$) | PCR($\downarrow$) | Acc | PCR | Acc | PCR | |
| ✓ | ✗ | ✗ | 80.10 (-10.96) | 18.37 (+11.17) | 91.06 | 7.20 | 87.57 | 11.38 | $1.81 \times 10^8$ |
| ✗ | ✓ | ✗ | 95.21 (-2.08) | 2.97 (+2.12) | 97.29 | 0.85 | 95.83 | 3.03 | $2.33 \times 10^{10}$ |
| ✗ | ✗ | ✓ | 90.94 (-1.52) | 7.32 (+1.55) | 92.46 | 5.77 | 85.31 | 13.67 | $1.02 \times 10^9$ |
| ✓ | ✓ | ✗ | 96.20 (-0.83) | 1.96 (+0.85) | 97.03 | 1.11 | 97.77 | 1.06 | $2.34 \times 10^{10}$ |
| ✓ | ✗ | ✓ | 92.24 (-2.44) | 5.99 (+2.48) | 94.68 | 3.51 | 95.73 | 3.13 | $1.20 \times 10^9$ |
| ✗ | ✓ | ✓ | 97.79 (+0.33) | 0.34 (-0.33) | 97.46 | 0.67 | 96.78 | 2.06 | $2.43 \times 10^{10}$ |
| ✓ | ✓ | ✓ | 98.12 (—) | / | 98.12 | / | 98.82 | / | $2.45 \times 10^{10}$ |
| Average | | | 92.94 (-2.50) | 6.16 (+2.98) | 95.44 | 3.18 | 93.98 | 5.72 | / |

We posit that this phenomenon stems from an implicit bias within multimodal image understanding models (hereafter "**multimodal models**"), which tend to favor certain "preferred" modalities during training (Peng et al. (2022b)). This biased optimization allows dominant modalities to disproportionately influence the gradient updates, leading to their features being well-optimized while those of other modalities are neglected. Such imbalance results in disparate levels of feature learning across the encoders, causing significant performance fluctuations when the model is faced with different combinations of inputs at inference time. This leads to the first pivotal question of our research: **How can we identify and quantify the modality preferences inherent in a multimodal model?**

While existing work has attempted to improve robustness by balancing modality contributions (Wei et al. (2023); Yang et al. (2024c)), they typically operate in the spatial domain and, as our experiments show, have not yet reached their performance ceiling. We argue that a critical, yet overlooked, source of information lies in the frequency domain. The distinct textural and structural features of different modalities manifest as unique signatures in the frequency spectrum, offering a more robust basis for comparison. Experimental analysis reveals a crucial insight: models are predominantly reliant on low-frequency information for decision-making (Fig. 1). Therefore, diverging from existing works (Peng et al. (2022b); Yang et al. (2024d)), we introduce the **F**requency **R**atio **M**etric (FRM), a novel indicator to *evaluate modality preference*. As shown in Table 1, our findings confirm a strong correlation: modalities with a higher FRM tend to be the ones that dominate the training process.

Building upon the FRM, we introduce **M**ultimodal **W**eight **A**llocation **M**odule (MWAM), which is a plug-and-play module designed to *actively mitigate modality bias*. By assigning weights inversely proportional to the FRM, MWAM actively rebalances the model's focus in gradient or loss space, steering the model's optimization trajectory towards a more equitable and robust representation.

The principal contributions of this paper are as follows: *Firstly*, we establish through experimental and theoretical validation that the dominance relationship between modalities can be effectively discerned and quantified in the frequency domain. Based on this finding, we introduce a new metric for this purpose, termed FRM. *Secondly*, we develop a novel plug-and-play module based on the FRM, termed MWAM, to dynamically allocate modal weights within each mini-batch. MWAM fosters more robust feature learning and, in contrast to existing gradient balancing methods, is inherently scalable to multiple modalities and maintains a simple backpropagation path. *Finally*, we conduct experiments across diverse modality combinations, various visual understanding tasks, and different backbone architectures to demonstrate the method's effectiveness and generality.

## 2 RELATED WORK

### 2.1 INCOMPLETE MULTIMODAL LEARNING

Enhancing the robustness of multimodal models in the presence of missing modalities has emerged as a critical research direction. Contemporary efforts in this domain can be broadly categorized into two dominant paradigms: imputation-based and imputation-free methods (Zhang et al. (2024a)).

**Imputation-based methods** attempt to reconstruct the missing data. These strategies range from transforming raw data into kernel or graph representations for imputation (Liu et al. (2017; 2020); Liu (2024); Wen et al. (2021)) to employing model-based approaches that recover modality information from the original data (Tran et al. (2017); Lin et al. (2023)). However, these methods often necessitate auxiliary recovery modules, introducing significant computational overhead that renders them unviable for deployment on resource-constrained devices.

In contrast, **imputation-free methods** operate exclusively on available modalities, forgoing reconstruction entirely (Zhang et al. (2024a); Reza et al. (2024); Lee et al. (2023)). While these approaches are computationally efficient and have fewer parameters, they often suffer from performance degradation, as they cannot compensate for the information lost from the absent modality.

A more promising direction, which has recently gained traction, **explores the latent relationships between modalities** to dynamically assess their contributions (Li et al. (2023b); Du et al. (2023); Wei et al. (2023); Zhang et al. (2024b)). Building upon this principle, we leverage frequency-domain analysis to probe the latent contributions of different modalities. This allows us to introduce the FRM that measures a model's intrinsic preference for each modality. Furthermore, the proposed MWAM serves as a potent balancing mechanism, dynamically modulating the contributions of different model components to provide more balanced attention to different modalities.

## 2.2 FREQUENCY DOMAIN DEEP LEARNING

Convolution in the frequency domain offers a distinct advantage over its spatial counterpart by preserving global feature integrity during iterative extraction and mitigating information loss. This inherent efficiency and representational power have spurred the increasing prominence of deep learning architectures that operate directly in the frequency domain.

In image processing, frequency information characterizes the rate of change in pixel intensities, with low frequencies corresponding to smooth regions and high frequencies to edges and textures. Historically, many models treated the frequency-based module as an auxiliary branch, using it to impose priors or constraints on the primary spatial backbone network (Yang et al. (2023; 2024a); Guo et al. (2023)). More recently, the trend has shifted towards a deeper integration where models are designed to directly leverage frequency-domain insights. For instance, (Xu et al. (2020)) developed a learning-based frequency selection method that prunes irrelevant frequency components without compromising accuracy, thereby boosting performance in image understanding tasks.

Building on the critical finding that models exhibit distinct preferences for low- and high-frequency channels (Xu et al. (2020); Wang et al. (2023); He et al. (2024)), we investigate approaches to mitigate performance degradation caused by missing modalities. We posit that these frequency preferences are a key indicator of a model's intrinsic bias toward specific modalities, and that this understanding can be leveraged to enhance the robustness of multimodal models.

## 3 FREQUENCY-DOMAIN MANIFESTATIONS OF MODALITY PREFERENCE

### 3.1 EXPLORATORY ANALYSIS OF FREQUENCY COMPONENTS

Our core insight is that the preference of multimodal models for specific modalities can be observed and quantified in the frequency domain. To this end, we first conduct an exploratory analysis to investigate how different frequency components impact multimodal learning. Specifically, we employ the Fast Fourier Transform (FFT) to decompose images into their frequency representations. We then apply a series of low-pass and high-pass filters with varying cutoff frequencies (i.e., window sizes) to generate frequency-filtered datasets for model training. The resulting learning dynamics, illustrated by the training loss and accuracy curves in Fig. 1, reveal three key insights.

*Firstly*, low-frequency components are crucial for mitigating training bias. As shown in Fig. 1-a, configurations that preserve low frequencies exhibit lower training losses. *Secondly*, low-frequency information facilitates sustained optimization. Models trained exclusively on high-frequency content exhibit early loss saturation (around 30 epochs), whereas models exposed to low-frequency bands continue to learn, as indicated by their dynamic loss curves (Fig. 1-b). *Finally*, while low-frequency components are dominant in determining final performance, high-frequency infor-

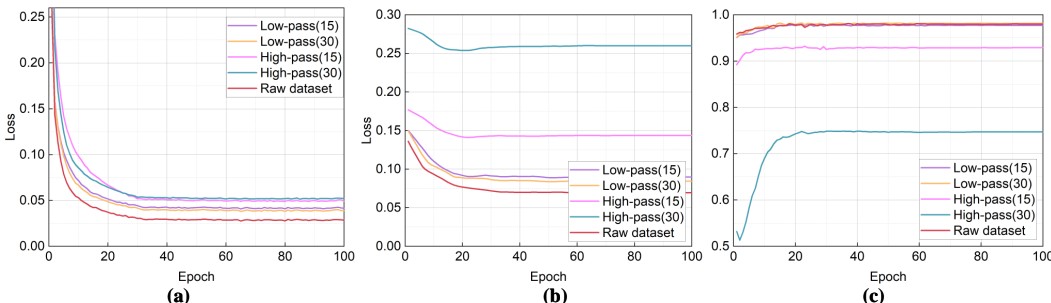

Figure 1: Impact of different frequency components on multimodal model performance. **(a):** Training loss curves. **(b):** Validation loss curves. **(c):** Validation accuracy curves. The numbers in the legend represent the window sizes of the filters, and "**Raw dataset**" denotes the original dataset without any filtering applied. More experimental details can be found in **Appendix A.5**.

mation provides a non-negligible contribution. Fig. 1-c demonstrates that incorporating more high-frequency content consistently boosts model accuracy.

These findings establish a principle for our motivation: an effective modality preference metric must prioritize the foundational signal from low frequencies while strategically incorporating the fine-grained details provided by high frequencies.

## 3.2 FREQUENCY-DOMAIN PREFERENCES IN MULTIMODAL MODELS

While Section 3.1 empirically outlined the core design principles of FRM based on experimental observations, this section aims to uncover the underlying reasons driving the frequency-domain preferences of multimodal models. Our goal is to provide substantial theoretical grounding for our proposed methods.

**Preliminaries.** Consider a training set $\mathcal{D} = \{x_{m_1}^j, x_{m_2}^j, y^j\}_{j=1,\ldots,N}$ for a dual-modal $K$-way classification task. Let $f(E_{m_1}, E_{m_2}, C)$ denote the complete classification network employing a concatenation-based fusion strategy, where $E_{m_1}(\cdot; \theta_{m_1})$ and $E_{m_2}(\cdot; \theta_{m_2})$ represent the modality-specific encoders, and $C(\cdot; W, b)$ serves as the classifier. The model is optimized using the gradient descent algorithm with Cross-Entropy loss.

**Theorem 3.1.** Let the training set, model architecture, loss function, and optimizer be defined as in the **Preliminaries**. The update rule for each modal branch is coupled through a shared global error signal. A modality significantly superior at the start of training will dominate the back-propagated error signals due to effective loss reduction, thereby hampering the optimization of weaker modalities through gradient suppression.

The detailed proof of Theorem 3.1 can be found in the Appendix A.1.1. This theorem confirms that the model does not attend to modalities equally but rather establishes a preference hierarchy during optimization. Building on this fact, we proceed to map this preference relationship into the frequency domain. This allows us to characterize why the model prefers certain modalities and observe the spectral signatures of this optimization imbalance.

**Theorem 3.2.** For a neural network with sufficient width, the training dynamics along the eigenvectors of the Neural Tangent Kernel (NTK) are decoupled. The convergence rate of the loss function along the $i$-th eigenvector direction is strictly determined by its corresponding eigenvalue $\lambda_i$:

$$\text{Decay Rate} \propto (1 - \eta \lambda_i), \tag{1}$$

where $\eta$ denotes the learning rate of the model.

The detailed proof of Theorem 3.2 can be found in the Appendix A.1.2. This theorem elucidates an intrinsic property of neural network optimization: eigenvectors associated with larger eigenvalues exhibit faster convergence rates. Moreover, extensive prior research has substantiated that these eigenvectors correspond to low-frequency functions within the input data (Rahaman et al. (2019); Wang et al. (2021); Basri et al. (2019)). Synthesizing these findings with Theorem 3.1, we arrive at a critical conclusion: during the optimization process, multimodal models exhibit a preferential

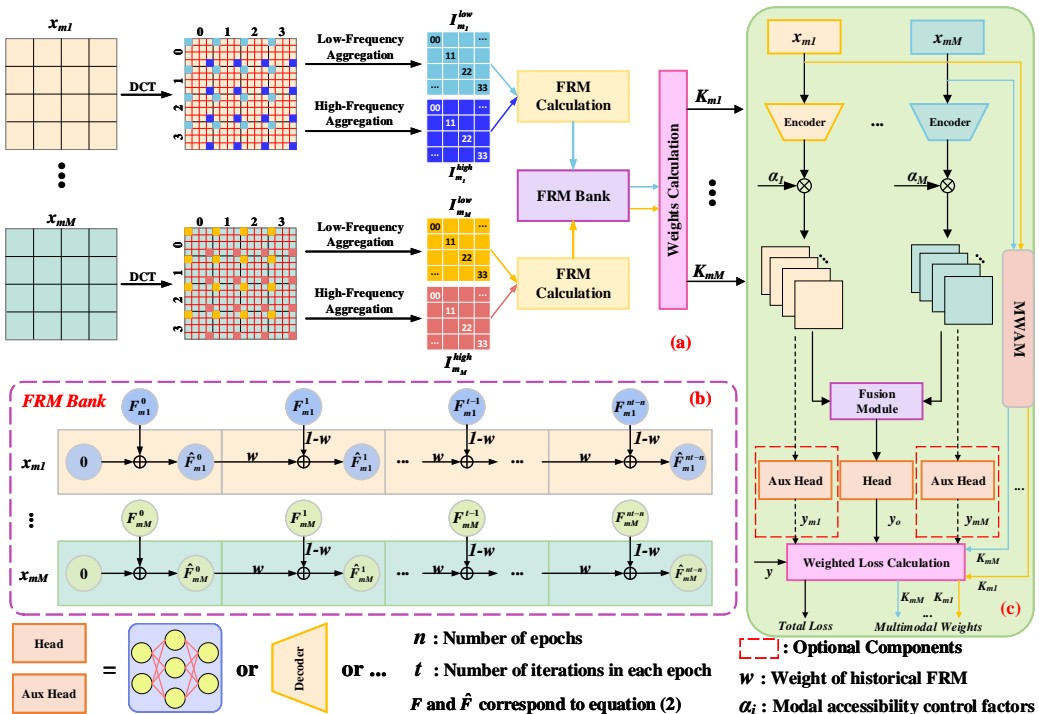

Figure 2: Architecture and application of our proposed MWAM. **(a):** Main structure of the MWAM. **(b):** FRM bank, designed to handle modality exceptions. Its update mechanism is governed by Eq. 2. **(c):** An illustration of the integration of MWAM into a multimodal host model. The calculation rules of FRM follow Eq. 4, which requires flipping and aligning the high-frequency components.

bias towards modalities rich in low-frequency information. Consequently, the persistence of this preference severely inhibits the optimization of the weaker modality branch.

To mitigate this reliance on frequency-specific preferences, we design the MWAM to first quantify the extent of the model's bias using FRM towards different modalities. Subsequently, by assigning adaptive weight coefficients to each modality, the MWAM effectively alleviates modality imbalance, thereby ensuring a balanced optimization process.

## 4 PROPOSED METHOD

### 4.1 MULTIMODAL WEIGHT ASSIGNMENT MODULE

We design the MWAM, a plug-and-play module that dynamically allocates guidance weights to each modal branch of the multimodal host model during training, thus providing a more equitable optimization environment for the model, as shown in Fig. 2. As illustrated, MWAM is composed of two key components: **(a)** a core structure responsible for extracting frequency-domain features and computing our FRM, and **(b)** an FRM bank designed to handle anomalous inputs. This section details the overall architecture of MWAM, while the design of the FRM is elaborated in Section 4.2.

The workflow of MWAM begins by separating the input into a grid of non-overlapping $p \times p$ patches. The Discrete Cosine Transform (DCT) is then applied to each patch, converting it into the frequency domain. From each transformed patch, we extract the top-left $q \times q$ block as the low-frequency component and the bottom-right $q \times q$ block as the high-frequency component. These frequency components are then spatially reassembled from their respective patch locations to construct two distinct feature maps representing the frequency profile of the entire input. We set the patch size $p = 8$ and the frequency block size $q = 2$ in this paper, unless mentioned otherwise. The resulting FRM for each modality is then processed by its corresponding FRM bank, which employs an update

rule (detailed in Fig. 2-b) to smooth the metric and derive the final guidance weights, formulated as:

$$\hat{F}_{m_i}^j = \begin{cases} F_{m_i}^j & j = 0 \\ \omega \hat{F}_{m_i}^{j-1} + (1 - \omega)F_{m_i}^j & j = 1, .., nt - n \end{cases}, \tag{2}$$

where $F_{m_i}^j$ is the FRM of $x_{m_i}^j$, $x_{m_i}^j$ is the $j$-th mini-batch of modality $m_i$, $\hat{F}_{m_i}^j$ is the FRM output from the $j$-th iteration FRM bank, which integrates the past state and the current state, and $\omega$ is the weight of historical FRM. We set $\omega = 0.5$, unless mentioned otherwise.

## 4.2 MODALITY PREFERENCE METRIC

**Motivation:** Identifying the dominant modality is a crucial step toward mitigating the robustness degradation caused by imbalanced model optimization. Prevailing strategies typically assess this imbalance in the spatial domain at the feature level (Peng et al. (2022b); Wei et al. (2024); Huang et al. (2025); Wei et al. (2023)), thereby overlooking crucial frequency-domain characteristics that offer a global and independent perspective. Motivated by the Frequency Principle of neural networks (Xu et al. (2024)) and our analysis in Section 3.1, we hypothesize that modality preference is quantifiable in the frequency domain. To this end, we introduce a novel metric, termed FRM, designed to diagnose this preference in real-time.

**Frequency Ratio Metric:** In Section 3.1, we present a key insight: model predictions are predominantly influenced by low-frequency components. This induces a learning bias, leading the model to favor modalities rich in low-frequency information, which typically encode fundamental structural and textural patterns. Based on this observation, an intuitive way to define modality preference is by the overall magnitude of the low-frequency components, quantified as their $L_1$-norms:

$$MP(x_{m_i}) = \sum_{a=0}^{w-1} \sum_{b=0}^{h-1} |I_{m_i}^{low}(a,b)|, \tag{3}$$

where $I_{m_i}^{low}(a,b)$ is the low-frequency components of the $x_{m_i}$, $w = Wq/p$ and $h = Hq/p$ are the width and height of the $I_{m_i}^{low}$ respectively. Here, $W$ and $H$ are the width and height of the $x_{m_i}$.

However, we further reveal that high-frequency components still provide valuable discriminative cues for the model. This finding suggests that a strategy of completely discarding these components, as implemented in Eq. 3, is inherently suboptimal. To address this limitation, we formally define the preference metric as the $L_1$-norm of the ratio between the low- and high-frequency components:

$$FRM(x_{m_i}) = \sum_{a=0}^{w-1} \sum_{b=0}^{h-1} |\frac{I_{m_i}^{low}(a,b)}{I_{m_i}^{high}(w-1-a, h-1-b) + \sigma}|, \tag{4}$$

where $I_{m_i}^{high}(a,b)$ is the high-frequency components of the $x_{m_i}$, $\sigma$ is a scaling factor.

In Eq. 4, when $I_{m_i}^{high}(a,b) = 0$, $\sigma$ will scale up the action of $I_{m_i}^{low}$ at the corresponding position. In addition, on modalities where low-frequency energies are similar, Eq. 4 effectively amplifies the FRM differences between these modalities, thereby increasing the weight gap and enhancing the method's performance. To validate our design choice, we evaluate several alternative FRM design rules in **Appendix A.8**.

## 4.3 HOW TO USE FRM FOR TRAINING INTERVENTION?

Altering a model's optimization trajectory via methods like gradient editing and weighted loss functions is a well-established paradigm for controlling model behavior during training (Peng et al. (2022b); Yang et al. (2024d); Huang et al. (2025)). We leverage this principle by first employing FRM to quantify the model's reliance on each modality. Based on this measurement, we intervene in the training process through two distinct mechanisms: gradient editing and dynamic loss weighting (see Fig. 3). Crucially, our gradient editing approach is parameter-free, and the dynamic loss weighting method necessitates the introduction of lightweight auxiliary heads with negligible parameter cost. To implement this, we define a dynamic weight allocation function, which assigns adaptive weights to each modality within every mini-batch. The function can be expressed as:

$$K_{m_i}^j(x_{m_i}^j) = \alpha - \frac{\beta}{1 + e^{-\lambda(T-\gamma)}}, \tag{5}$$

where $\alpha$, $\beta$, $\lambda$, and $\gamma$ are adjustable scaling factors. $T$ is the proportion of FRM of $x_{m_i}^j$ to the mean FRM of all modalities in that mini-batch, denoted as:

$$T = \frac{FRM(x_{m_i}^j)}{\frac{1}{M}\sum_{c=1}^{M} FRM(x_{m_c}^j) + \sigma}, \tag{6}$$

Eq. 6 dynamically assesses the contribution of each modality within a mini-batch, informing the assignment of corresponding weights during gradient updates. A key characteristic of Eq. 5 is its inherent flexibility; it is designed to extend beyond dual-modality scenarios to accommodate an arbitrary number of modalities. In **Appendix A.9**, we conducted detailed ablation experiments on these scaling factors in order to show the sensitivity of these factors.

Integrating the MWAM into a multimodal host model can significantly enhance its training fairness (as shown in Fig. 2-c), with the corresponding pseudocode provided in **Appendix A.2**. For models that compute modality-specific losses, typically via auxiliary heads, the MWAM globally regulates the training process by weighting these losses, as formulated in the following equation:

$$\mathcal{L}_{total} = \sum_{i=1}^{M} K_{m_i}\mathcal{L}(y, y_{m_i}) + \mathcal{L}(y, y_o), \tag{7}$$

where $y$, $y_o$, and $y_{m_i}$ are the ground-truth label, the total output of the model, and the output of the auxiliary head, respectively. $K_{m_i}$ can be calculated from Eq. 5 and Eq. 6. **It is worth noting that** the auxiliary heads and weighted loss are optional extensions rather than required, allowing for flexible integration.

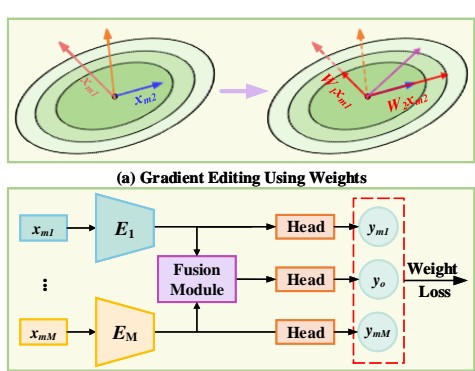

(a) Gradient Editing Using Weights

(b) Weighted Loss Using Auxiliary Heads and Weights

Figure 3: Schematic of training intervention mechanisms. **(a)** is parameter-free, and **(b)** needs extra lightweight auxiliary heads.

In its base configuration, MWAM is entirely parameter-free, incurring only a negligible computational overhead in terms of floating-point operations during training.

## 5 EXPERIMENTS

To validate the efficacy and generalizability of our method, we conduct extensive experiments on a diverse range of multimodal visual understanding tasks. In this section, we highlight the principal results, while comprehensive experimental details and further analyses are deferred to the **Appendix**.

### 5.1 EXPERIMENTS ON MULTIMODAL SEGMENTATION

**Datasets & Metrics:** We validate our method on two challenging segmentation tasks: brain tumor segmentation and semantic segmentation. For brain tumor segmentation, we utilize the BRATS2020 dataset (Menze et al. (2015)). We split datasets into 219, 50, and 100 subjects for training, validation, and testing, respectively. For semantic segmentation, we use the NYU-Depth V2 dataset (Nathan Silberman & Fergus (2012)). We split the dataset into 795 images for training and 654 images for testing, adhering to a standard 40-class label setting following official recommendations. Model performance is evaluated using the widely recognized Dice coefficient of the whole tumor for brain tumor segmentation and Mean Intersection over Union (MIoU) for semantic segmentation.

**Baselines:** We select HeMIS (Havaei et al. (2016)), Robust Segmentation (Chen et al. (2020)), RFNet (Ding et al. (2021)), mmFormer (Zhang et al. (2022)), A2FSeg (Wang & Hong (2023)), M3AE (Liu et al. (2023)), GSS (Qiu et al. (2023)), and LS3M (Zhang et al. (2025)) as comparison models for the brain tumor segmentation, and ESANet-MD (Seichter et al. (2021)) and MMANet

Table 2: Performance comparison of multimodal robust solutions on BRATS2020 dataset. † denotes the integration of our proposed MWAM with the corresponding base model. **Bold** and *italics* indicate the **best value** and the *second best value* for each row in turn. Due to space constraints, we present only a subset of the comparison results here. More results are provided in **Table 15**, which includes several advanced methods, including HeMIS, Robust Seg, M3AE, LS3M, and A2FSeg.

| Modal | | | | RFNet | | RFNet† | | mmFormer | | mmFormer† | | GSS | | GSS† | |
|---|---|---|---|---|---|---|---|---|---|---|---|---|---|---|---|
| F | T1 | T1c | T2 | Dice(↑) | PCR(↓) | Dice | PCR | Dice | PCR | Dice | PCR | Dice | PCR | Dice | PCR |
| ✗ | ✗ | ✗ | ✓ | 85.07 | 5.62 | *85.93* | 5.32 | 84.71 | 5.91 | 85.80 | **4.96** | 85.78 | 5.64 | **86.57** | *5.23* |
| ✗ | ✗ | ✓ | ✗ | 74.95 | 16.85 | *76.39* | 15.83 | 74.92 | 16.78 | 76.00 | *15.82* | 75.96 | 16.44 | **79.08** | **13.43** |
| ✗ | ✓ | ✗ | ✗ | 71.98 | 20.15 | 74.12 | 18.33 | 74.56 | 17.18 | *75.93* | *15.89* | 75.38 | 17.08 | **79.03** | **13.49** |
| ✓ | ✗ | ✗ | ✗ | 85.16 | 5.52 | 86.49 | 4.70 | 85.87 | 4.62 | *86.75* | **3.91** | 85.99 | 5.41 | **87.34** | *4.39* |
| ✗ | ✗ | ✓ | ✓ | 86.91 | 3.58 | *87.86* | *3.20* | 86.80 | 3.59 | 87.54 | **3.04** | 87.39 | 3.87 | **88.40** | 3.23 |
| ✗ | ✓ | ✓ | ✗ | 78.48 | 12.94 | 80.14 | 11.70 | 79.48 | 11.72 | 80.09 | *11.29* | *80.30* | 11.67 | **81.93** | **10.31** |
| ✓ | ✓ | ✗ | ✗ | 88.08 | 2.29 | *89.06* | 1.87 | 88.42 | 1.79 | 88.75 | *1.69* | 88.87 | 2.24 | **90.05** | **1.42** |
| ✗ | ✓ | ✗ | ✓ | 86.94 | 3.55 | *87.89* | *3.16* | 86.96 | 3.41 | 87.52 | **3.06** | 87.75 | 3.48 | **88.11** | 3.55 |
| ✓ | ✗ | ✗ | ✓ | 88.70 | 1.60 | 89.27 | 1.64 | 88.61 | 1.58 | *89.31* | **1.07** | 89.02 | 2.08 | **90.07** | *1.40* |
| ✓ | ✗ | ✓ | ✗ | 88.46 | 1.86 | 89.49 | 1.40 | 88.62 | 1.57 | 89.27 | **1.12** | *89.88* | *1.13* | 89.91 | 1.58 |
| ✓ | ✓ | ✓ | ✗ | 89.36 | 0.87 | 89.84 | 1.01 | 89.15 | 0.98 | 89.45 | 0.92 | *90.20* | *0.78* | **90.82** | **0.58** |
| ✓ | ✓ | ✗ | ✓ | 89.74 | 0.44 | *90.46* | **0.33** | 89.55 | 0.53 | 89.93 | *0.39* | 90.23 | 0.75 | **90.80** | 0.60 |
| ✓ | ✗ | ✓ | ✓ | 89.44 | 0.78 | 89.99 | 0.85 | 89.54 | 0.54 | 89.98 | **0.33** | *90.36* | 0.60 | **91.02** | *0.36* |
| ✗ | ✓ | ✓ | ✓ | 87.74 | 2.66 | *88.50* | 2.49 | 87.51 | 2.80 | 88.03 | **2.49** | 88.14 | 3.05 | **88.97** | *2.61* |
| ✓ | ✓ | ✓ | ✓ | 90.14 | / | 90.76 | / | 90.03 | / | 90.28 | / | *90.91* | / | **91.35** | / |
| Average | | | | 85.41 | 5.62 | *86.41* | 5.13 | 85.64 | 5.21 | 86.30 | *4.71* | *86.41* | 5.30 | **87.56** | **4.44** |

(Wei et al. (2023)) for the semantic segmentation. The ESANet is a well-established real-time RGB-D segmentation method widely recognized in academia and industry. We adapt it with modality-access control factors to create ESANet-MD, which handles incomplete modality inputs.

**Implementation Details:** We embed MWAM into RFNet, mmFormer, and GSS for brain tumor segmentation, and into ESANet-MD and MMANet for semantic segmentation. Due to the design of the baseline architectures, we do not introduce additional auxiliary heads. We utilize the official code provided by the baseline and comply with the experimental configuration of the respective underlying model. For both tasks, we set the scaling factors in Eq. 5 to 1.5, 1, 6, and 0.7.

**Results:** Results are shown in Tables 2 and 3. In Table 2, the integration of MWAM consistently enhances the performance of all host methods, leading them to outperform their vanilla models. Significant improvements are observed in both Dice and PCR metrics. Notably, when embedded with MWAM, earlier methods such as RFNet and mmFormer achieve an average Dice score comparable to the SOTA method LS3M, and even surpass it in terms of PCR. Furthermore, GSS, when equipped with MWAM, outperforms LS3M on both Dice and PCR. This suggests that our MWAM enables the model to identify a more optimal decision boundary. It is important to highlight that RFNet, mmFormer, and GSS are all SOTA methods designed for the multimodal missing modality problem. Our MWAM further elevates their performance ceiling. Furthermore, the fact that mmFormer is a ViT-based model underscores the adaptability of our method to diverse architectures.

To further demonstrate MWAM's compatibility with various fusion strategies, we conduct semantic segmentation experiments by integrating it into the vanilla ESANet-MD. ESANet-MD is known for performing feature fusion during the extraction stage, and our results confirm that MWAM seamlessly adapts to this architecture. Both methods enhanced by MWAM show distinct performance gains in MIoU and PCR. In summary, our MWAM not only exhibits strong compatibility with diverse architectures and fusion strategies but also provides substantial performance boosts to existing robust methods, enabling them to push beyond their intrinsic performance limitations.

## 5.2 EXPERIMENTS ON MULTIMODAL CLASSIFICATION

**Datasets & Metrics:** We utilize CASIA-SURF (Zhang et al. (2019)) for conducting our experiments in the multimodal classification task. We adapt the intra-testing protocol recommended by the vanilla work, dividing it into train, validation, and test sets with 29k, 1k, and 57k samples, respectively. We

Table 3: Performance comparison of multimodal robust solutions on NYU-Depth V2 dataset

| Modal | | ESANet-MD | | MMANet | | ESANet-MD† | | MMANet† | |
|---|---|---|---|---|---|---|---|---|---|
| Rgb | Depth | MIoU(↑) | PCR(↓) | MIoU | PCR | MIoU | PCR | MIoU | PCR |
| ✓ | ✗ | 41.30 | 13.07 | 42.24 | 12.47 | *44.27* | *9.28* | **45.88** | **8.77** |
| ✗ | ✓ | 38.79 | 18.35 | 39.93 | *17.26* | *40.34* | 17.34 | **41.64** | **16.55** |
| ✓ | ✓ | 47.51 | / | 48.26 | / | *48.80* | / | **49.90** | / |
| **Average** | | 42.53 | 15.71 | 43.47 | 14.87 | *44.47* | *13.31* | **45.81** | **12.66** |

Table 4: Performance comparison of multimodal robust solutions on SURF dataset. Due to space constraints, we present only a subset of the comparison results here. Comprehensive results are provided in **Appendix A.14**.

| Modal | | | SF-MD | | MMFormer | | CRMT-JT | | MMANet | | SF-MD† | | MMANet† | |
|---|---|---|---|---|---|---|---|---|---|---|---|---|---|---|
| Rgb | Depth | Ir | Acc(↑) | PCR(↓) | Acc | PCR | Acc | PCR | Acc | PCR | Acc | PCR | Acc | PCR |
| ✓ | ✗ | ✗ | 83.92 | 13.83 | 88.85 | 9.39 | 90.13 | 8.66 | 91.43 | 7.77 | *92.13* | *6.87* | **93.49** | **5.32** |
| ✗ | ✓ | ✗ | 93.65 | 3.84 | 95.33 | 2.78 | 96.26 | 2.45 | *97.73* | *1.41* | 96.55 | 2.41 | **98.31** | **0.44** |
| ✗ | ✗ | ✓ | 89.60 | 8.00 | 86.01 | 12.29 | 89.75 | 9.05 | 89.96 | 9.25 | *93.76* | *5.23* | **94.18** | **4.62** |
| ✓ | ✓ | ✗ | 95.43 | 2.01 | 98.07 | -0.01 | 95.68 | 3.04 | *98.39* | 0.75 | 98.25 | 0.69 | **98.64** | 0.10 |
| ✓ | ✗ | ✓ | 93.26 | 4.24 | 95.23 | 2.89 | 91.81 | 6.96 | **96.99** | **2.16** | *96.62* | 2.33 | 96.60 | *2.17* |
| ✗ | ✓ | ✓ | 96.74 | 0.67 | 96.90 | 1.18 | 98.74 | -0.06 | *98.82* | 0.31 | 98.63 | 0.30 | **99.27** | -0.54 |
| ✓ | ✓ | ✓ | 97.39 | / | 98.06 | / | 98.68 | / | **99.13** | / | *98.93* | / | 98.74 | / |
| **Average** | | | 92.85 | 5.43 | 94.07 | 4.75 | 94.44 | 5.02 | 96.06 | 3.61 | *96.41* | *2.97* | **97.03** | **2.02** |

use Acc to quantify models' classification performance, and employ PCR to measure the extent of performance degradation when the model uses incomplete inputs for inference.

**Baselines:** We compare our method to a list of SOTA methods, including SF-MD, HeMIS (Havaei et al. (2016)), RFNet (Ding et al. (2021)), mmFormer (Zhang et al. (2022)), CRMT (Yang et al. (2024c)), COM (Qian & Wang (2023)), DCP (Hu et al. (2024)), and MMANet (Wei et al. (2023)). SF (Zhang et al. (2019)) is the official network for the SURF, while SF-MD is a variant specifically designed to simulate missing modality scenarios by employing modality access control factors.

**Implementation Details:** We integrate MWAM into SF-MD and MMANet to demonstrate the performance gains that MWAM can bring to the host model. In SF-MD, we additionally introduce auxiliary heads, whereas we do not in MMANet. We follow the experimental settings recommended by MMANet, including epochs, mini-batch size, etc. Additionally, we set the scaling factors in Eq. 5 to 1.5, 1, 6, and 0.7, respectively. All experiments were conducted on an NVIDIA RTX 3090 GPU.

**Results:** The results are shown in Table 4. After incorporating MWAM, the foundational SF-MD model demonstrates a significant boost in both average accuracy and PCR. Notably, the accuracy for the RGB modality, the weakest in single-modality scenarios, surged by 8.21%. This not only indicates that the vanilla SF-MD possesses untapped performance potential but also proves that MWAM can empower simple, original methods to overcome performance bottlenecks and enhance their robustness. Furthermore, when embedded to MMANet, an already high-performing contemporary method for missing modalities, MWAM pushes its capabilities even further. Although it yields sub-optimal results on certain modality combinations, the MWAM-enhanced MMANet achieves the highest average score, which can be attributed to the model's balanced attention across different modalities during training. Remarkably, the performance of SF-MD, when augmented by MWAM, even surpasses that of recent SOTA methods such as mmFormer and CRMT-JT. In summary, these results collectively demonstrate that MWAM not only delivers substantial performance gains to base models but also enables existing robust models to break through their own inherent performance ceilings. More comparison results can be found in **Appendix**.

## 5.3 ABLATION STUDIES

We conduct a comprehensive suite of **ablation studies and parameter sensitivity analyses**. We present an analysis here on the performance impact of our intervention mechanism due to space constraints. The complete experimental results are detailed in the **Appendix A.6 to A.10**.

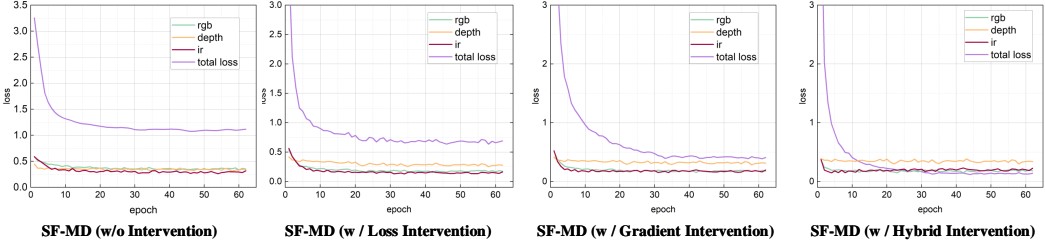

Figure 4: Training losses for different interventions

Table 5: Impact of constraints at different levels on model performance metrics. We use SF-MD as the baseline framework and conduct experiments on the SURF dataset.

| Modal | | | w/o MWAM | | Loss | | Gradient | | Hybrid Loss+Grad | |
| --- | --- | --- | --- | --- | --- | --- | --- | --- | --- | --- |
| Rgb | Depth | Ir | Acc | PCR | Acc | PCR | Acc | PCR | Acc | PCR |
| ✓ | ✗ | ✗ | 83.92 | 13.83 | *89.42* | *9.37* | 89.36 | 9.67 | **92.13** | **6.87** |
| ✗ | ✓ | ✗ | 93.65 | 3.84 | 95.12 | *3.60* | *95.27* | 3.70 | **96.55** | **2.41** |
| ✗ | ✗ | ✓ | 89.60 | 8.00 | 91.25 | 7.52 | **94.26** | **4.72** | *93.76* | *5.23* |
| ✓ | ✓ | ✗ | 95.43 | 2.01 | *97.18* | *1.51* | 97.06 | 1.89 | **98.25** | **0.69** |
| ✓ | ✗ | ✓ | 93.26 | 4.24 | 95.72 | 2.99 | *96.61* | *2.35* | **96.62** | **2.33** |
| ✗ | ✓ | ✓ | 96.74 | 0.67 | 98.05 | 0.63 | **98.64** | **0.29** | *98.63* | *0.30* |
| ✓ | ✓ | ✓ | 97.39 | / | *98.67* | / | **98.93** | / | **98.93** | / |
| Average | | | 92.85 | 5.43 | 95.06 | 4.27 | *95.73* | *3.77* | **96.41** | **2.97** |

The quantitative results in Table 5 confirm that all intervention strategies improve performance over the baseline. Gradient-level intervention yields superior gains compared to loss-level intervention. We hypothesize this is because manipulating losses alone can disrupt global training dynamics and introduce subtle optimization trade-offs in the fusion block. In contrast, a hybrid approach proves most effective, achieving a synergistic balance between global and local modality attention.

Fig. 4 provides a visualization of these training dynamics. The baseline model exhibits entangled loss curves for each modality, indicating a competitive and unbalanced optimization process. In contrast, embedding MWAM stratifies the modality losses: the Depth modality, having the highest FRM, consistently maintains the highest loss, while the other modalities cluster together at lower loss values. This stratification shows that MWAM effectively guides the model to prioritize learning from modalities with lower FRM. Furthermore, the overall training process becomes more stable, as evidenced by the reduced variance in the total loss curve. This suggests that MWAM fosters a more balanced and synergistic learning environment, enabling the model to harmonize contributions from all inputs, overcome performance bottlenecks, and ultimately achieve superior performance.

## 6 CONCLUSION

We propose a simple yet efficient method to enhance the robustness of multimodal models from a frequency-domain perspective. Our approach is motivated by the common observation that models often develop a learning bias towards easier modalities, leading to under-optimization of other modality-specific pathways. To counteract this imbalance, we introduce two key contributions: (1) the **F**requency **R**atio **M**etric (FRM), a novel indicator to quantify this modality preference, and (2) the **M**ultimodal **W**eight **A**llocation **M**odule (MWAM), a plug-and-play component that uses FRM to dynamically modulate the influence of each modality in the gradient or loss space. This training intervention mechanism enforces a more balanced learning dynamic, enabling existing multimodal architectures to break through performance ceilings with minimal overhead, and to enhance their robustness. We conducted experiments across diverse modality combinations, various visual understanding tasks, and different backbone architectures to demonstrate the method's effectiveness and generality. Additionally, we discuss the limitations of our method in **Appendix A.14**.

## LLM USAGE STATEMENT

In this work, we only employed a Large Language Model (LLM) as an auxiliary tool to polish the manuscript. Specifically, its use was focused on enhancing the clarity and fluency of the writing to better articulate our core insights and findings.

## REPRODUCIBILITY STATEMENT

To ensure reproducibility, the hyperparameters and environmental configurations for our main experiments are detailed in the manuscript, specifically within Sections 5.1 and 5.2.

## IMPACT STATEMENT

This paper presents work whose goal is to advance the field of Machine Learning. There are many potential societal consequences of our work, none which we feel must be specifically highlighted here.

## ACKNOWLEDGMENT

This work was supported in part by the National Natural Science Foundation of China under Grant 62273353.

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

# A APPENDIX

We provide theoretical rationale for the design of MWAM and FRM, a more detailed experimental implementation, and a richer description of the experimental content in the supplementary materials to provide reviewers with additional reference information.

In summary, this appendix is organized as follows:

- **Section A.1** provides the theoretical derivations underpinning the design of our FRM and MWAM.
- **Section A.2** presents the pseudocode for our proposed algorithm.
- **Section A.3** offers further details on the datasets and experimental implementation.
- **Section A.4** details the evaluation metrics used for each task.
- **Section A.5** provides supplementary details and analysis for the experiments presented in Section 3.1 of the main paper.
- **Section A.6** includes an analysis of the computational overhead introduced by our method.
- **Section A.7** presents a sensitivity analysis on the size of the frequency sampling window in MWAM.
- **Section A.8** investigates the impact of different design choices for the FRM on model performance.
- **Section A.9** reports the hyperparameter sensitivity analysis for the weighting function (Eq. 5).
- **Section A.10** evaluates the performance of MWAM under different training schemes, namely small-batch and online learning settings.
- **Section A.11** showcases performance on the action recognition task using video and optical flow as input modalities.
- **Section A.12** highlights the performance gains achieved by applying MWAM to a baseline model for multimodal object detection.
- **Section A.13** provides a comparative analysis of MWAM against state-of-the-art methods for multimodal imbalance optimization.
- **Section A.14** discuss the limitations of MWAM and potential avenues for future work.
- **Section A.15** presents an extended comparison with a wider range of methods addressing missing modalities, which was omitted from the main paper due to space constraints.
- **Section A.16** demonstrates the effectiveness of MWAM on the task of fine-grained visual classification.

## A.1 PROOF OF THE THEOREM

In this section, we provide detailed proofs for the two theorems in Section 3.2 to enhance the completeness of the manuscript.

We adopt the notations established in the Preliminaries (Section 3.2). Specifically, we address a $K$-way classification task involving two modalities, denoted as $x_{m_1}$ and $x_{m_2}$.

**Definition 1**. Let the training dataset be defined as a collection of $N$ samples:

$$\mathcal{D} = \{x^j, y^j\}_{j=1,2,..,N}, \tag{8}$$

where each input $x^j = \{x^j_{m_1}, x^j_{m_2}\}$ consists of data from two modalities, and $y^j \in \{1, 2, \ldots, K\}$ represents the corresponding ground-truth label.

**Definition 2.** We define the multimodal classification model as a composite function:

$$f(E_{m_1}, E_{m_2}, C), \tag{9}$$

where $E_{m_1}(\cdot; \theta_{m_1})$ and $E_{m_2}(\cdot; \theta_{m_2})$ denote the feature encoders for the respective modalities, parameterized by $\theta_{m_1}$ and $\theta_{m_2}$. $C(\cdot; W, b)$ represents the classifier head parameterized by weights $W$ and bias $b$.

**Definition 3.** The model is optimized by minimizing the empirical risk via the Cross-Entropy loss, formulated as:

$$L = -\frac{1}{N} \sum_{j=1}^{N} \log \frac{e^{f(x^j)_{y^j}}}{\sum_{q=1}^{K} e^{f(x^j)_q}}. \tag{10}$$

**Assumption 1.** To facilitate the analysis of gradient behavior, we consider a canonical fusion approach based on feature concatenation. Under this setting, the network output $f(x^j)$ is expressed as:

$$\begin{aligned}
f(x^j) &= W \cdot [E_{m_1}(x^j_{m_1}; \theta_{m_1}) \oplus E_{m_2}(x^j_{m_2}; \theta_{m_2})] + b \\
&= W_{m_1} E_{m_1}(x^j_{m_1}; \theta_{m_1}) + W_{m_2} E_{m_2}(x^j_{m_2}; \theta_{m_2}) + b,
\end{aligned} \tag{11}$$

where $\oplus$ denotes the concatenation operation.

### A.1.1 PROOF OF THEOREM 3.1

Under the assumptions outlined above, when optimized using the gradient descent algorithm, the optimization procedure proceeds as follows:

For the parameters specific to modality $m_1$ (with analogous steps applying to modality $m_2$), the update rules at iteration $t$ are reformulated as:

$$W^{t+1}_{m_1} = W^t_{m_1} - \eta \nabla_{W_{m_1}} L(W^t_{m_1}) = W^t_{m_1} - \eta \frac{1}{N} \sum_{j=1}^{N} \frac{\partial L}{\partial f(x^j)} E_{m_1}(x^j_{m_1}, \theta_{m_1}), \tag{12}$$

$$\theta^{t+1}_{m_1} = \theta^t_{m_1} - \eta \nabla_{\theta_{m_1}} L(\theta^t_{m_1}) = \theta^t_{m_1} - \eta \frac{1}{N} \sum_{j=1}^{N} \frac{\partial L}{\partial f(x^j)} \frac{\partial (W^t_{m_1} E_{m_1})}{\partial \theta^t_{m_1}}, \tag{13}$$

where $\eta$ denotes the learning rate. The gradient of the loss with respect to the network output $f(x^j)$ is derived as:

$$\frac{\partial L}{\partial f(x^j)} = \frac{e^{(W_{m_1} E_{m_1}(x^j_{m_1}; \theta_{m_1}) + W_{m_2} E_{m_2}(x^j_{m_2}; \theta_{m_2}) + b)_c}}{\sum_{q=1}^{K} e^{(W_{m_1} E_{m_1}(x^j_{m_1}; \theta_{m_1}) + W_{m_2} E_{m_2}(x^j_{m_2}; \theta_{m_2}) + b)_q}} - 1|_{c=y^j}, \tag{14}$$

Eq. 12 and Eq. 13 indicate that the multimodal branches are optimized independently, with the sole exception of the coupling term $\frac{\partial L}{\partial f(x^j)}$. This implies that the parameter updates of one modality are not directly interfered with by the other, aside from this shared error signal.

Within the term $\frac{\partial L}{\partial f(x^j)}$, the contributions of different modalities are aggregated via dot products. Consequently, Eq. 14 can be reformulated in a unified vector notation as:

$$\frac{\partial L}{\partial f(x^j)} = \frac{e^{(W \cdot E^T + b)_c}}{\sum_{q=1}^{K} e^{(W \cdot E^T + b)_q}} - 1|_{c=y^j}, \tag{15}$$

where $W = [W_{m_1}, W_{m_2}]$, $E = [E_{m_1}, E_{m_2}]$ represent the concatenated weights and embeddings, respectively, and superscript $T$ denotes the matrix transpose. The convergence behavior of the gradients can be analyzed via the eigendecomposition of $E$.

Eq. 15 reveals a critical insight: if one modal branch performs significantly better, it will dominate the shared gradient signal during training. Consequently, the optimization process becomes biased, effectively establishing that branch as the model's preferred modality.

### A.1.2 PROOF OF THEOREM 3.2

**Definition 4.** To facilitate the theoretical analysis, without loss of generality, we define a simplified setting involving a single modality $m_1$ over the dataset $\{(x^j_{m_1}, y^j)\}_{j=1}^{N}$. Let $f(x, \theta)$ denote the neural network, where $\theta \in \mathbb{R}^n$ represents the learnable parameters and $x \in \mathbb{R}^p$ is the input. The optimization objective is defined as the minimization of the squared error loss function $L(\theta)$:

$$L(\theta) = \frac{1}{2} \sum_{j=1}^{N} \left( f(x^j_{m_1}, \theta) - y^j \right)^2, \tag{16}$$

where the parameter sequence $\{\theta_t\}$ is generated via gradient descent with a learning rate $\eta$, governed by the update rule:

$$\theta_{t+1} = \theta_t - \eta\nabla L(\theta_t), \tag{17}$$

**Dynamics Analysis.** Let $q_t = (f(x_{m_1}^j, \theta))_{j\in[N]} \in \mathbb{R}^N$ be the network output vector at iteration $t$, and $y = (y^j)_{j\in[N]} \in \mathbb{R}^N$ be the target vector. Following prior works on the linearization of wide neural networks (Dong et al. (2019); Du et al. (2019); Oymak & Soltanolkotabi (2018)), the evolution of the residual (prediction error) can be approximated by a quasi-linear dynamics:

$$q_{t+1} - y \approx (I - \eta\hat{H}_t)(q_t - y), \tag{18}$$

where $\hat{H}_t$ is the empirical Gram matrix defined as:

$$\hat{H}_t = \left(\left\langle \int_0^1 \frac{\partial f(x_{m_1}^j, \theta_t + \alpha(\theta_{t+1} - \theta_t))}{\partial\theta}d\alpha, \frac{\partial f(x_{m_1}^i, \theta_t)}{\partial\theta} \right\rangle\right)_{j,i}. \tag{19}$$

Crucially, as the network width $d \to \infty$, it has been established that $\hat{H}_t$ converges to a static matrix $H^*$, known as the Neural Tangent Kernel (NTK) (Jacot et al. (2020); Oymak & Soltanolkotabi (2018)). Specifically, $\|\hat{H}_t - H^*\|_F = o(1)$. Thus, in the infinite-width limit, we have $\hat{H}_t \approx H^*$, where $H^*$ is a positive semi-definite $N \times N$ matrix.

Let $\lambda_1 \geq \lambda_2 \geq ... \geq \lambda_n \geq 0$ denote the eigenvalues of $H^*$, and let $u_1, u_2, ..., u_N$ be the corresponding orthonormal eigenvectors. The spectral decomposition of $H^*$ is given by:

$$H^* = \sum_{i=1}^N \lambda_i u_i u_i^T. \tag{20}$$

We now analyze the optimization trajectory by projecting the error signal onto the eigenvector basis. The dynamics in the direction of the $i$-th eigenvector $u_i$ are:

$$\langle q_{t+1} - y, u_i\rangle = \langle (I - \eta H^*)(q_t - y), u_i\rangle = (1 - \eta\lambda_i)\langle q_t - y, u_i\rangle. \tag{21}$$

Eq. 21 demonstrates that the convergence rate along each eigen-direction is determined by the factor $(1 - \eta\lambda_i)$. Consequently, components corresponding to larger eigenvalues $\lambda_i$ decay significantly faster.

## A.2 PSEUDO-CODE FOR THE ALGORITHM

We continue the assumption of Section A.1 that $\mathcal{D} = \{x^j, y^j\}_{j=1,2,..,N}$, $f(E_{m_1}, E_{m_2}, C)$, et al. To make it easier to express, we discard here the bias in $C$. Therefore, Algorithm 1 demonstrates the training procedure for the underlying model + MWAM.

## A.3 MORE DETAILED EXPERIMENTAL IMPLEMENTATION

We provide supplementary details on the implementation of each experiment according to the sequence outlined in the main text.

We conducted multimodal classification experiments on CASIA-SURF. CASIA-SURF is a large-scale benchmark dataset for face anti-spoofing, consisting of 87k sets of RGB, Depth, and IR images. We integrated MWAM into the SF-MD and MMANet models, resulting in SF-MD+MWAM and MMANet+MWAM. We followed the experimental settings recommended for MMANet, specifically: we set the mini-batch size to 64 and trained the models for 100 epochs using the SGD optimizer. For the SGD optimizer, weight decay and momentum were set to 0.0005 and 0.9, respectively. The initial learning rate was set to 1e-3 with a learning rate decay strategy: in SF-MD+MWAM, the learning rate was divided by 10 every 30 epochs, while in MMANet+MWAM, it was divided by 10 at epochs 16, 33, and 50. Additionally, the parameters $(\alpha, \beta)$ in MMANet were set to (30, 0.5). All images were resized to $112\times112$ pixels before being fed into the network.

We conducted multimodal semantic segmentation experiments on NYU-Depth V2. NYU-Depth V2 is a large-scale dataset for indoor scene semantic segmentation with 1,449 images containing

---

**Algorithm 1** Training of underlying model+MWAM.

---

**Input**: $\mathcal{D} = \{x_{m_1}^j, x_{m_2}^j, y^j\}_{j=1}^N$, iterations: $T_s$, learning rate: $\eta$, and other input parameters in baseline.

1: **for** $t = 0, 1, ..., T_s - 1$ **do**
2:     Sample a mini-batch $B^t = \{x_{m_1}, x_{m_2}, y\}$;
3:     Extract frequency components $I_{m_1(m_2)}^{low(high)}$ using MWAM;
4:     Calculate $FRM(x_{m_1})$ and $FRM(x_{m_2})$ using Eq. 4;
5:     Update $FRMs$ using Eq. 2, and calculate weights $K_{m_1}$ and $K_{m_2}$ using Eq. 5 and Eq. 6;
6:     Feed-forward the batched data $B^t$ to the model;
7:     Calculate the loss using Eq. 7, and compute the gradients $g(\theta_{m_1}^t)$, $g(\theta_{m_2}^t)$ and $g(\theta_C^t)$ using backpropagation;
8:     Update the gradients according to the following rules:
    $\theta_{m_1}^{t+1} \leftarrow \theta_{m_1}^t - K_{m_1}(\eta \cdot g(\theta_{m_1}^t))$
    $\theta_{m_2}^{t+1} \leftarrow \theta_{m_2}^t - K_{m_2}(\eta \cdot g(\theta_{m_2}^t))$
    $\theta_C^{t+1} \leftarrow \theta_C^t - \eta \cdot g(\theta_C^t)$
9: **end for**

---

RGB and Depth modalities. We integrated MWAM into the ESANet-MD and MMANet models, resulting in improved versions: ESANet-MD+MWAM and MMANet+MWAM. We followed the experimental settings recommended for MMANet: the mini-batch size was set to 8, and the models were trained for 300 epochs using the SGD optimizer. For the SGD optimizer, weight decay and momentum were set to 1e-4 and 0.9, respectively. The initial learning rate was set to 1e-2, and PyTorch's one-cycle scheduler was used. Additionally, the parameters $(\alpha, \beta)$ in MMANet were set to (4, 0.2). All images were resized to 480×640 pixels before being fed into the network.

We conducted multimodal brain tumor segmentation experiments on BRATS2020 which used for the MICCAI Brain Tumor Segmentation Challenge, includes 369 training subjects with 4 modalities each. We integrated MWAM into the RFNet and mmFormer models, resulting in improved versions: RFNet+MWAM and mmFormer+MWAM. For model training, we set the mini-batch size to 2 and used the Adam optimizer ($\beta_1 = 0.9$, $\beta_2 = 0.999$), training for 300 epochs and 1000 epochs, respectively. The weight decay for the Adam optimizer was set to 1e-5 and 1e-4 for the two models, respectively, and both used the "poly" learning rate policy. All images were resized to 80×80×80 pixels before being fed into the network.

## A.4 EVALUATION METRICS

We use Accuracy (Acc) to quantify the model's performance on multimodal binary classification tasks, Mean Intersection over Union (MIoU) to measure the model's performance on multimodal semantic segmentation tasks, and the Dice coefficient for the whole tumor to evaluate the model's performance on multimodal brain tumor segmentation tasks. In addition, we use the Performance Collapse Rate (PCR) to assess the performance collapse for different incomplete modality combinations across various tasks.

We conducted a multimodal binary classification task in the body of our text; therefore, the accuracy discussed in this paper should be binary accuracy. Binary accuracy is mathematically represented as follows:

$$\text{Acc} = \frac{\text{TP} + \text{TN}}{\text{TP} + \text{TN} + \text{FP} + \text{FN}} \tag{22}$$

where TP, TN, FP, and FN represent True Positives, True Negatives, False Positives, and False Negatives, respectively.

Intersection over Union (IoU) is a metric used to evaluate model performance in image segmentation tasks. MIoU is the average of IoU values across categories, commonly used to assess multi-class segmentation tasks. Mathematically, the MIoU calculation formula is as follows:

$$\text{MIoU} = \frac{1}{N} \sum_{i=1}^N \text{IoU}_i \tag{23}$$

where:

$$\text{IoU}_i = \frac{\text{TP}_i}{\text{TP}_i + \text{FP}_i + \text{FN}_i} \tag{24}$$

where $\text{TP}_i$, $\text{FP}_i$, and $\text{FN}_i$ represent True Positives, False Positives, and False Negatives for class $i$, respectively. Higher MIoU values indicate better performance in multimodal semantic segmentation

The Dice coefficient (also known as the Dice similarity coefficient) is a metric used to measure the similarity between two sample sets and is one of the most commonly used metrics for brain tumor image segmentation tasks. Its calculation formula is as follows:

$$\text{Dice}(\hat{y}, y) = \frac{2 \cdot \|\hat{y}, y\|_1}{\|\hat{y}\|_1 + \|y\|_1} \tag{25}$$

where $y$ and $\hat{y}$ represent ground truth and predictions, respectively. Larger Dice scores indicate that the predictions are more similar to the ground truth, thus reflecting better segmentation accuracy.

Furthermore, we use the PCR to quantify model performance degradation under incomplete modality inputs. PCR is defined as the rate of performance decline when using various combinations of incomplete modality inputs compared to complete modality inputs, mathematically represented as:

$$PCR = \frac{A(MM(x_{full}, \theta)) - A(MM(x_{miss}, \theta))}{A(MM(x_{full}, \theta))} \tag{26}$$

where $A(\cdot)$ represents evaluation metrics for different multimodal tasks, $MM(\cdot, \theta)$ is a multimodal model parameterized by $\theta$, $x_{full}$ denotes the complete modality input, and $x_{miss}$ denotes different combinations of incomplete modality inputs. Generally, a lower PCR value indicates that the impact of missing modality inputs on model performance is smaller, meaning the performance is closer to that with complete modality inputs.

### A.5 MORE SUPPLEMENTARY NOTES ON FREQUENCY DIFFERENCE EXPERIMENTS

In Section 3.1, in order to illustrate that different frequency bands in the frequency domain have different levels of influence in the multimodal visual comprehension model, we used different types of filters with different window sizes to process the dataset post-training, and obtained the corresponding results and conclusions. However, due to space constraints, we present more detailed experiments here.

We first use FFT processing in the spatial domain for the three modalities RGB, Depth and IR, and move the DC component to the center of the image. After that, a filter of size $n \times n$ is designed with the image center as the reference. Among them, the low-pass filter retains only $n \times n$ pixel points in the image center, and the other positions are set to 0. The high-pass filter discards $n \times n$ pixel points in the image center. Finally, the iFFT process is performed using the filtered spectrogram to obtain the filtered spatial domain image, and the results of each stage of the operation are shown in Figure 5.

In Figure 5, images of different modalities present different visual effects in the spatial domain, e.g., the RGB image contains richer textures and details, and the Depth map contains more reflective of the structural hierarchy, despite the loss of these textures and detailed information. However, when these images are converted to the frequency domain, the differences are more obvious, e.g., the energy of the Depth map is more concentrated in some small number of frequency bands, while the energy of the RGB and IR images is more scattered. In addition, the processed images with low-pass and high-pass filters having different window sizes exhibit different characteristics. For example, the high-pass filter aims to preserve the high-frequency components of the image, and the processed image retains the edge information in the image. Compared to a high-pass filter of size $30 \times 30$, a filter of size $15 \times 15$ retains more frequency components, and therefore it retains more edge information. The low-pass filter aims to retain the low-frequency components in the image, and the processed image retains the texture, detail features in the image. However, since the low frequency energy is more concentrated, the low pass filter size has a greater impact on the results. Low-pass filters using $15 \times 15$ retain most of the feature detail in the image, but they do not retain as much detail as the results of the $30 \times 30$ size filter. **It is also worth noting that we set the window sizes of the filters to 15 and 30 just to illustrate the general pattern found, and that there are no special values taken.**

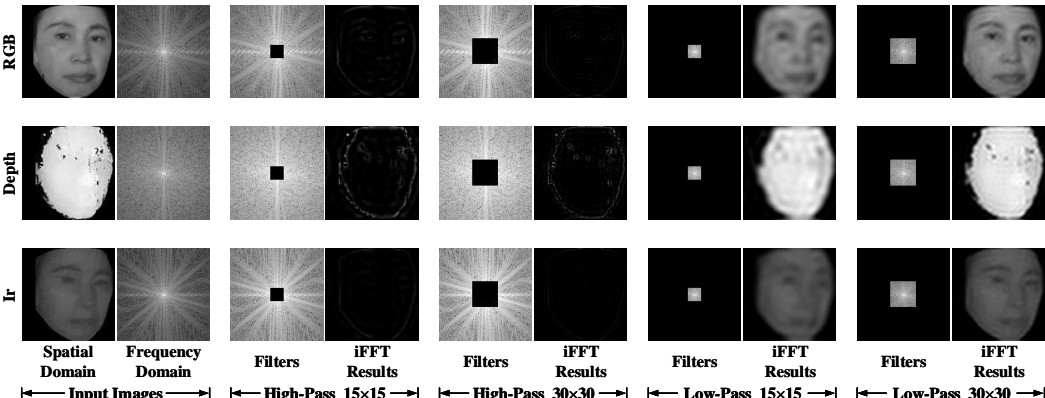

Figure 5: Comparison of visualization of filter effects. In the horizontal axis, it can be divided into five parts, which are the spatial and frequency domain images of the input image, two high-pass filter results with different window sizes, and two low-pass filter results with different window sizes.

Table 6: Computational cost analysis of MWAM. The **Parameters** and **FLOPs** statistics are reported for the entire model (with and without MWAM), while the **Inference Time** is measured for the MWAM module in isolation.

| Tasks | Classification | | Semantic Segmentation | | Brain Tumor Segmentation | |
|---|---|---|---|---|---|---|
| **Models** | MMANet | MMANet +MWAM | ESANet-MD | ESANet-MD +MWAM | RFNet | RFNet +MWAM |
| **Params. (M)** | 13.18 | 13.18 (+0) | 71.69 | 71.69 (+0) | 8.4 | 8.4 (+0) |
| **FLOPs (G)** | 60.63 | 60.6752 (+0.0452) | 526.60 | 526.6627 (+0.0627) | 204.56 | 204.5604 (+0.0004) |
| **Inference time** | 0.0385±0.005 | | 0.0878±0.0040 | | 0.0542±0.0139 | |

Therefore, the extent to which the model benefits from the different frequency bands of these images is different. There is reason to believe that multimodal preferences can be reasonably modeled from frequency bands.

### A.6 COMPUTATIONAL OVERHEAD OF MWAM

To assess the computational overhead of MWAM, we report the parameter count, FLOPs, and MWAM inference time for the three main experiments, both with and without the module. These results are detailed in Table 6.

The experimental results are highly promising. The computational overhead of MWAM, primarily attributed to DCT and FRM computations, is negligible. This allows it to yield significant performance gains for existing baselines at a minimal marginal cost, even for those that are already robust. Crucially, MWAM is engineered to refine the model's optimization trajectory exclusively during the training phase. The module is detached for inference, thereby incurring zero additional computational overhead at runtime.

### A.7 IMPACT OF FREQUENCY SAMPLING WINDOW SIZE ON MODEL PERFORMANCE

In MWAM, we sample the input frequency components using a $q \times q$ window from patches of size $p \times p$. In our work, we set $p = 8$ to align with most baseline methods. The size of the frequency sampling window $q$ directly affects the calculation range of FRM. As $q$ increases, more frequency points are included in the computation, impacting the output of MWAM. Therefore, we explore the effect of the sampling window size in MWAM.

Table 7: Impact of frequency selection window size on model performance. We embed MWAM into MMANet and vary the sampling window size $q$. In the table, **"w/o MWAM"** is vanilla MMANet.

| Modal | | | w/o MWAM | | q=1 | | q=2 | | q=4 | | q=6 | |
|---|---|---|---|---|---|---|---|---|---|---|---|---|
| Rgb | Depth | Ir | Acc(↑) | PCR(↓) | Acc | PCR | Acc | PCR | Acc | PCR | Acc | PCR |
| ✓ | ✗ | ✗ | 91.43 | 7.77 | *93.07* | *5.92* | **93.49** | **5.32** | 92.43 | 6.54 | 89.93 | 9.22 |
| ✗ | ✓ | ✗ | 97.73 | 1.41 | **98.62** | **0.31** | 98.31 | 0.44 | *98.61* | *0.29* | 98.51 | 0.56 |
| ✗ | ✗ | ✓ | 89.96 | 9.25 | 90.62 | 8.40 | **94.18** | **4.62** | *91.90* | *7.08* | 91.18 | 7.95 |
| ✓ | ✓ | ✗ | 98.39 | 0.75 | **98.86** | **0.07** | 98.64 | *0.10* | *98.68* | 0.22 | 98.00 | 1.07 |
| ✓ | ✗ | ✓ | **96.99** | **2.16** | 96.52 | 2.44 | *96.60* | *2.17* | 96.19 | 2.74 | 95.71 | 3.38 |
| ✗ | ✓ | ✓ | 98.82 | 0.31 | *99.18* | -0.25 | **99.27** | -0.54 | 98.65 | 0.25 | 99.14 | -0.08 |
| ✓ | ✓ | ✓ | **99.13** | / | 98.93 | / | 98.74 | / | 98.90 | / | *99.06* | / |
| **Average** | | | 96.06 | 3.61 | *96.54* | *2.82* | **97.03** | **2.02** | 96.48 | 2.85 | 95.93 | 3.68 |

Table 8: Comparison of the impact of FRM calculation rules on model performance. In the table, **"w/o MWAM"** is vanilla MMANet, **"w/o High-Freq."** is in the form of an $L_1$-Norm sum using the low-frequency components, as in Eq. 3, **"w/ High-Freq."** is using our FRM as in Eq.4, **"Direct Sum"** is a directly summed $L_1$-Norm of the low-frequency and high-frequency components, as shown in Eq. 27, and **"Weighted Sum"** is the L1-Norm weighted sum of the low-frequency and high-frequency components, as shown in Eq. 28. All experiments were conducted in the same environment.

| Modal | | | w/o MWAM | | w/o High-Freq. | | w/ High-Freq. | | Direct Sum | | Weighted Sum | |
|---|---|---|---|---|---|---|---|---|---|---|---|---|
| Rgb | Depth | Ir | Acc(↑) | PCR(↓) | Acc | PCR | Acc | PCR | Acc | PCR | Acc | PCR |
| ✓ | ✗ | ✗ | 91.43 | 7.77 | 90.79 | 7.54 | **93.49** | **5.32** | *92.28* | *6.53* | 91.97 | 7.29 |
| ✗ | ✓ | ✗ | 97.73 | 1.41 | **98.38** | -0.19 | *98.31* | 0.44 | 97.36 | 1.39 | 98.30 | 0.91 |
| ✗ | ✗ | ✓ | 89.96 | 9.25 | 88.73 | 9.63 | **94.18** | **4.62** | 90.67 | *8.16* | *90.68* | 8.59 |
| ✓ | ✓ | ✗ | 98.39 | 0.75 | 98.21 | -0.02 | *98.64* | 0.10 | 98.58 | 0.15 | **98.87** | 0.33 |
| ✓ | ✗ | ✓ | **96.99** | *2.16* | 92.28 | 6.02 | *96.60* | 2.17 | 95.18 | 3.60 | 95.80 | 3.43 |
| ✗ | ✓ | ✓ | 98.82 | 0.31 | 98.42 | -0.23 | **99.27** | -0.54 | *98.98* | -0.25 | 98.83 | 0.37 |
| ✓ | ✓ | ✓ | *99.13* | / | 98.19 | / | 98.74 | / | 98.73 | / | **99.20** | / |
| **Average** | | | 96.06 | 3.61 | 95.00 | 3.79 | **97.03** | **2.02** | 95.96 | *3.26* | *96.23* | 3.49 |

Table 7 shows the performance differences of MMANet+MWAM when only the sampling window size $q$ is changed under the same experimental configuration. As $q$ increases, more frequency components are included in the FRM calculation, leading to a noticeable performance improvement. However, this enhancement has an upper limit. When the patch size $p$ is fixed, increasing the window size beyond a certain point leads to performance stagnation or even negative optimization. In our experiments, when $q \geq 4$, the experimental results of MMANet+MWAM show a degradation compared to MMANet. This degradation is not random. We believe this occurs because, at $q = 4$, irrelevant information from the mid-frequency components interferes with the FRM calculation. At $q = 6$, the sampled frequency components experience a certain degree of aliasing, leading to errors in the FRM calculation. As a result, this causes a deviation in the training direction of the model when guided by MWAM. This also demonstrates the effectiveness of our proposed method from another perspective. By intervening in the model's optimization path, it can surpass its existing performance levels.

## A.8 IMPACT OF FRM CALCULATION RULES ON MODULE PERFORMANCE

In section 4.2, we introduced two methods for measuring multimodal preference, differing based on the inclusion of high-frequency components in the FRM calculation. To evaluate these methods, we tested the multimodal classification performance of MMANet+MWAM on the SURF dataset under identical experimental settings. The results are presented in Table 8.

The results are consistent with our hypothesis. Excluding high-frequency components from the preference measurement leads to a noticeable decline in experimental performance due to the loss of crucial information, resulting in suboptimal outcomes. Specifically, performance drops by 1.06% com-

pared to the standard MMANet and by 2.03% compared to the FRM-based measurement method. This suggests that measuring multimodal model preferences using the FRM with high-frequency components (w/ High-Freq.) provides a more substantial performance advantage than employing the $L_1$-Norm with low-frequency components (w/o High-Freq.). This finding supports our hypothesis outlined in Section 4.2, which posits that high-frequency components in multimodal image understanding tasks have a measurable impact on model performance, although this impact is less pronounced than the performance improvements achieved through low-frequency components.

Furthermore, we explored two additional methods for measuring multimodal preferences that consider both low-frequency and high-frequency components. Unlike FRM, these methods use a frequency component summation approach.

Firstly, we define the measurement method as the sum of the $L_1$-Norms of the low-frequency and high-frequency components, which can be expressed as:

$$MP(x_{m_i}) = \sum_{a=0}^{w-1} \sum_{b=0}^{h-1} (|I_{m_i}^{low}(a,b)| + |I_{m_i}^{high}(a,b)|) \tag{27}$$

where $w$ and $h$ represent the width and height of the frequency features, respectively, while $I_{m_i}^{low}(a,b)$ and $I_{m_i}^{high}(a,b)$ denote the low-frequency and high-frequency components of $x_{m_i}$.

Additionally, to highlight the varying importance of different frequency bands for multimodal image understanding tasks, we use a weighted sum of the different frequency bands, which can be expressed as:

$$MP(x_{m_i}) = \sum_{a=0}^{w-1} \sum_{b=0}^{h-1} (\omega|I_{m_i}^{low}(a,b)| + (1-\omega)|I_{m_i}^{high}(a,b)|) \tag{28}$$

where $\omega$ is a weight factor for different frequency bands. In our experiment, we set $\omega = 0.9$.

The experimental results are shown in columns 5 (Direct Sum) and 6 (Weighted Sum) of Table 8. First, it is undeniable that the performance of the experimental groups that fully considered both low-frequency and high-frequency components was better than that of the experimental groups that completely discarded high-frequency components. This further proves that our approach of retaining high-frequency components is valid. Second, directly summing the $L_1$-Norms of the frequency components leads to a significant performance drop. This is because such a measurement scheme blurs the energy boundaries between frequencies, making the impacts of high-frequency and low-frequency components similar, thus preventing the model from properly allocating weights based on low-frequency components. Furthermore, after applying weights to different frequency bands, the situation improves significantly. However, applying a fixed weight factor is still suboptimal because the sporadic zeros in high frequencies can alter the preference measurement calculation (sum calculations aggregate all energy). Therefore, we use a ratio-based approach, like FRM, which employs proportional factors to address the low-frequency components at positions corresponding to zeros in high-frequency features.

Therefore, we choose FRM as the metric for measuring multimodal preferences and use it to optimize the model.

### A.9 PARAMETER SENSITIVITY ANALYSIS FOR THE WEIGHT ALLOCATION FUNCTION

Fundamentally, Eq. 5 represents a variant of the sigmoid function that has been horizontally reflected to create a decreasing curve. Its behavior is governed by four hyperparameters that enable independent scaling and translation along both the $x$ and $y$ axes. We use this function to process the output of Eq. 6, which quantifies the relationship between a single modality's FRM and the mean FRM across all modalities, constraining the input value, $T$, to the range [0, 2].

However, directly applying a standard or a simple reflected sigmoid for weight allocation presents two significant challenges. First, within the input range of [0, 2], the standard sigmoid function exhibits low sensitivity, meaning its output values are clustered within a narrow band. This results in minimal differentiation between the weights assigned to various modalities, thereby failing to effectively modulate the model's focus. Second, the sigmoid's output range of (0, 1) inherently suppresses the contribution of every modality during optimization. Although the degree of suppression varies, this universal dampening effect can lead to model underfitting, a scenario where overall performance is sacrificed for the sake of enforcing training balance.

Table 9: Parameter sensitivity analysis for the weight allocation function.

| Parameter Combination ($\alpha$ / $\beta$ / $\gamma$ / $\lambda$) | Average Accuracy (Acc) | Performance Collapse Rate (PCR) |
|---|---|---|
| 1.2 / 1.0 / 0.7 / 6.0 | 96.48 | 3.34 |
| 1.8 / 1.0 / 0.7 / 6.0 | 96.42 | 3.09 |
| 1.5 / 1.3 / 0.7 / 6.0 | 96.04 | 3.45 |
| 1.5 / 0.7 / 0.7 / 6.0 | 96.53 | *2.32* |
| 1.5 / 1.0 / 1.0 / 6.0 | 96.73 | 2.57 |
| 1.5 / 1.0 / 0.4 / 6.0 | 96.37 | 3.30 |
| 1.5 / 1.0 / 0.7 / 6.3 | 96.07 | 3.51 |
| 1.5 / 1.0 / 0.7 / 5.7 | 95.87 | 3.26 |
| 2.0 / 1.5 / 1.3 / 6.5 | **97.08** | 2.57 |
| 1.5 / 1.0 / 0.7 / 6.0 | *97.03* | **2.02** |

Table 10: Training models with different batch sizes. **Bold** indicates the best metric under the same batch size.

| Modal | | | bs=64 w/o MWAM | | bs=64 w/ MWAM | | bs=16 w/o MWAM | | bs=16 w/ MWAM | |
|---|---|---|---|---|---|---|---|---|---|---|
| Rgb | Depth | Ir | Acc($\uparrow$) | PCR($\downarrow$) | Acc | PCR | Acc | PCR | Acc | PCR |
| ✓ | ✗ | ✗ | 91.43 | 7.77 | **93.49** | **5.32** | 87.89 | 11.32 | **90.57** | **8.17** |
| ✗ | ✓ | ✗ | 97.73 | 1.41 | **98.31** | **0.44** | 97.83 | 1.29 | 97.36 | **1.29** |
| ✗ | ✗ | ✓ | 89.96 | 9.25 | **94.18** | **4.62** | 89.16 | 10.04 | **94.03** | **4.66** |
| ✓ | ✓ | ✗ | 98.39 | 0.75 | **98.64** | **0.10** | 97.01 | 2.12 | **98.34** | **0.29** |
| ✓ | ✗ | ✓ | 96.99 | 2.16 | 96.60 | 2.17 | **96.21** | 2.93 | 95.79 | **2.88** |
| ✗ | ✓ | ✓ | 98.82 | 0.31 | **99.27** | -0.54 | **98.65** | 0.46 | 98.29 | **0.34** |
| ✓ | ✓ | ✓ | **99.13** | / | 98.74 | / | **99.11** | / | 98.63 | / |
| **Average** | | | 96.06 | 3.61 | **97.03** | **2.02** | 95.12 | 4.69 | **96.14** | **2.94** |
| **Modal** | | | bs=4 w/o MWAM | | bs=4 w/ MWAM | | bs=1 w/o MWAM | | bs=1 w/ MWAM | |
| Rgb | Depth | Ir | Acc($\uparrow$) | PCR($\downarrow$) | Acc | PCR | Acc | PCR | Acc | PCR |
| ✓ | ✗ | ✗ | 87.10 | 11.80 | **89.94** | **9.31** | **76.66** | -17.74 | 72.02 | 4.69 |
| ✗ | ✓ | ✗ | 97.84 | 0.92 | **98.52** | **0.66** | **88.70** | -36.23 | 74.18 | 1.83 |
| ✗ | ✗ | ✓ | **94.21** | **4.60** | 93.38 | 5.84 | 74.71 | -14.74 | **80.98** | -7.17 |
| ✓ | ✓ | ✗ | 96.49 | 2.29 | **98.35** | **0.83** | 71.59 | -9.95 | **74.32** | 1.64 |
| ✓ | ✗ | ✓ | 95.63 | 3.16 | **96.51** | **2.68** | 75.16 | -15.44 | **75.55** | 0.01 |
| ✗ | ✓ | ✓ | 98.33 | 0.43 | **99.21** | -0.04 | 70.06 | -7.60 | **77.78** | -2.94 |
| ✓ | ✓ | ✓ | 98.75 | / | **99.17** | / | 65.11 | / | **75.56** | / |
| **Average** | | | 95.49 | 3.86 | **96.44** | **3.21** | 74.57 | -16.95 | **75.77** | -0.32 |

To overcome these limitations, we introduce the four aforementioned parameters to flexibly scale and translate the reflected sigmoid function. Recognizing that the disparity in FRM values across modalities varies significantly with different multimodal tasks and training paradigms, we formulate these parameters as tunable hyperparameters. This design choice enhances the generalizability and adaptability of our MWAM.

We conduct an additional parameter sensitivity analysis for Eq. 5 and Eq. 6, with the results presented in Table 9.

Experimental results demonstrate that the model's final performance is largely insensitive to these four adjustable scaling factors when they are within a reasonable range. However, excessively imbalanced settings—for instance, assigning weights to different modalities that span several orders of magnitude—can misguide the training process and lead to a collapse in overall performance.

## A.10 EVALUATING MWAM'S PERFORMANCE UNDER DIVERSE TRAINING SCENARIOS

Since MWAM dynamically adjusts modality weights on a batch-by-batch basis using spectral information, it is inevitably required to operate under diverse, and sometimes extreme, training scenarios.

Table 11: Performance comparison of video-optical flow multimodal classification tasks.

| Modal | | Baseline | | MWAM | | OGM-GE | | LFM | |
|---|---|---|---|---|---|---|---|---|---|
| Video | O.F. | Acc(↑) | PCR(↓) | Acc | PCR | Acc | PCR | Acc | PCR |
| ✓ | ✗ | **74.41** | **7.81** | 72.36 | 11.36 | 73.47 | *8.44* | *73.76* | 9.45 |
| ✗ | ✓ | 55.68 | 31.01 | **60.45** | **25.95** | 53.86 | 32.88 | *57.27* | *29.70* |
| ✓ | ✓ | 80.71 | / | **81.63** | / | 80.24 | / | *81.46* | / |
| Average | | 70.27 | *19.41* | **71.48** | **18.65** | 69.19 | 20.66 | *70.83* | 19.57 |

While our experiments in the main paper already cover a spectrum of batch sizes—from large (64 for classification) to small (8 for semantic segmentation and 2 for brain tumor segmentation)—we conduct a supplementary analysis here to further assess the robustness of MWAM. Specifically, we evaluate its performance on the classification task across a wider range of settings, including medium and small batches, as well as an extreme online learning scenario (i.e., a batch size of 1). The results are presented in Table 10.

As shown in the table, while the Acc and PCR metrics fluctuate with decreasing batch size, this volatility is primarily attributed to MMANet. In contrast, our MWAM maintains relatively stable performance. This stability stems from two key components: the scaling factor (Eq. 5) and the FRM Bank. The scaling factor allows for manual adjustments to adapt to diverse learning conditions, while the FRM Bank dynamically compensates for abnormal FRM values. This dual mechanism is particularly beneficial for small-batch learning. For instance, in the extreme cases of $bs = 4$ and $bs = 1$, we adjusted the hyperparameters to $\alpha = 1.5$, $\beta = 1$, $\gamma = 1$, and $\lambda = 4$, respectively.

### A.11 PERFORMANCE OF MWAM ON ACTION RECOGNITION

To further demonstrate the adaptability of MWAM to diverse modality combinations, we conducted an additional experiment on the action recognition task. This task utilizes video and optical flow as input modalities on the UCF-101 dataset. We benchmarked our approach against several modality-balancing baselines (Peng et al. (2022a); Yang et al. (2024b)), with the results summarized in Table 11.

Notably, we did not perform any task-specific hyperparameter tuning for this experiment, adhering to the default configuration reported in the main text. The results demonstrate that our MWAM enhances model performance in the absence of the dominant modality (video), thereby boosting its robustness. Furthermore, our method outperforms two other leading state-of-the-art approaches, demonstrating superior robustness in comparison.

### A.12 PERFORMANCE OF MWAM ON MULTIMODAL DETECTION

To further demonstrate the generalizability of MWAM to diverse multimodal tasks, we evaluate its performance on multimodal object detection using visible and infrared (RGB-IR) imagery.

To validate the effectiveness of MWAM, we integrated it into the Two-Stream YOLOv8n baseline and evaluated its detection performance on the DroneVehicle dataset. As detailed in Table 12, our module demonstrates a significant improvement over the baseline model. The visualization results of the recognition task and their corresponding PR curves are presented in Figs. 6 and 7, offering qualitative corroboration for the quantitative findings summarized in Table 12.

It should be noted that, a direct comparison with existing methods is not straightforward, as approaches for handling missing modalities or balancing multimodal inputs are not tailored for the multimodal detection task. Therefore, this section presents the experimental results for MWAM exclusively.

### A.13 COMPARISON WITH MORE COMPARATIVE METHODS

In addition to the methods in the text that focus on improving robustness to missing modalities, we provide a performance comparison of MWAM with advanced multimodal balanced training methods in recent years. These methods include: OGM-GE(Peng et al. (2022a)), LFM(Yang et al. (2024b)), AGM(Li et al. (2023a)), MSES and MSLR. We performed multimodal classification tasks under

Table 12: Performance comparison of MWAM in the multimodal detection task.

| Modal | | T-yolov8n | | T-yolov8n+MWAM | |
|---|---|---|---|---|---|
| RGB | IR | mAP50(↑) | PCR(↓) | mAP50 | PCR |
| ✓ | ✗ | 0.215 | 72.78 | **0.628** | **21.89** |
| ✗ | ✓ | 0.669 | 15.32 | **0.738** | **8.21** |
| ✓ | ✓ | 0.790 | / | **0.804** | / |
| **Average** | | 0.558 | 44.05 | **0.723** | **15.05** |

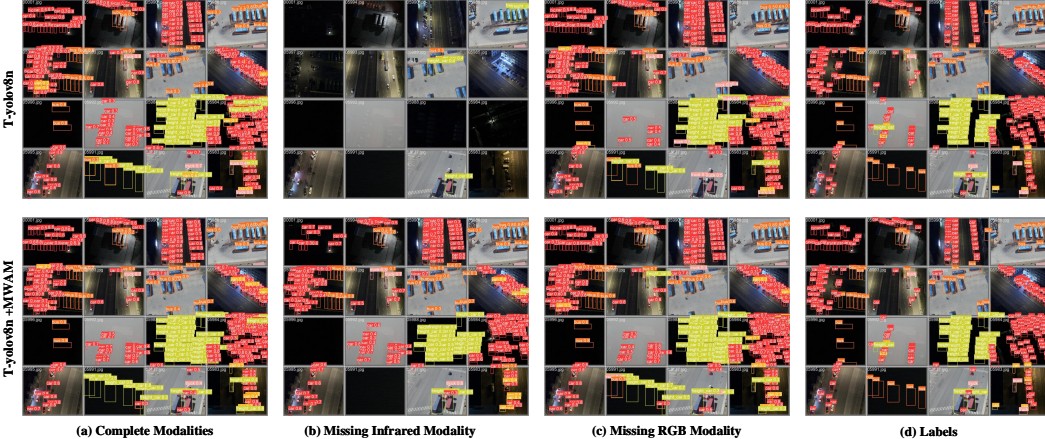

(a) Complete Modalities  (b) Missing Infrared Modality  (c) Missing RGB Modality  (d) Labels

Figure 6: Qualitative results for object detection.

the same experimental conditions. Due to width constraints, we split the comparison tables into Tables 13 and 14. Bold indicates the optimal metrics, and italic indicates the suboptimal ones. Our method achieves most optimal and some suboptimal results, showing it effectively enhances model robustness by balancing multimodal preferences.

Table 13: Comparison with different balanced multimodal training methods. To adapt to some comparison methods, we trained our MWAM using pairwise combinations of three modalities. All methods used the vanilla ResNet18 structure and concat-based fusion, and were tested on the SURF dataset.

| Modal | | OGM-GE | | LFM | | AGM | | MWAM | |
|---|---|---|---|---|---|---|---|---|---|
| Rgb | Depth | Acc(↑) | PCR(↓) | Acc | PCR | Acc | PCR | Acc | PCR |
| ✓ | ✗ | 79.87 | 17.43 | 69.29 | 29.67 | *88.81* | *9.86* | **93.99** | **4.96** |
| ✗ | ✓ | 87.49 | 9.55 | **98.81** | -0.29 | 94.42 | 4.16 | *97.68* | 1.22 |
| ✓ | ✓ | 96.73 | / | *98.52* | / | 98.52 | / | **98.89** | / |
| **Average** | | 88.03 | 13.49 | 88.78 | 14.69 | *93.92* | *7.01* | **96.85** | **3.09** |
| Rgb | Ir | Acc(↑) | PCR(↓) | Acc | PCR | Acc | PCR | Acc | PCR |
| ✓ | ✗ | 87.23 | 9.71 | *88.21* | **2.86** | 82.23 | 11.11 | **93.34** | *3.40* |
| ✗ | ✓ | *88.02* | 8.89 | 76.30 | 15.98 | 85.02 | *8.10* | **89.60** | 7.28 |
| ✓ | ✓ | *96.61* | / | 90.81 | / | 92.51 | / | **96.63** | / |
| **Average** | | *90.62* | *9.30* | 85.11 | 9.42 | 86.59 | 9.60 | **93.19** | **5.34** |
| Depth | Ir | Acc(↑) | PCR(↓) | Acc | PCR | Acc | PCR | Acc | PCR |
| ✓ | ✗ | 86.79 | 12.40 | *97.28* | *1.34* | 96.84 | 0.83 | **98.12** | **1.00** |
| ✗ | ✓ | 70.17 | 29.18 | 67.85 | 31.19 | 65.52 | 32.90 | **91.70** | **7.48** |
| ✓ | ✓ | *99.08* | / | 98.60 | / | 97.65 | / | **99.11** | / |
| **Average** | | 85.35 | 20.79 | *87.91* | *16.26* | 86.67 | 16.87 | **96.31** | **4.24** |

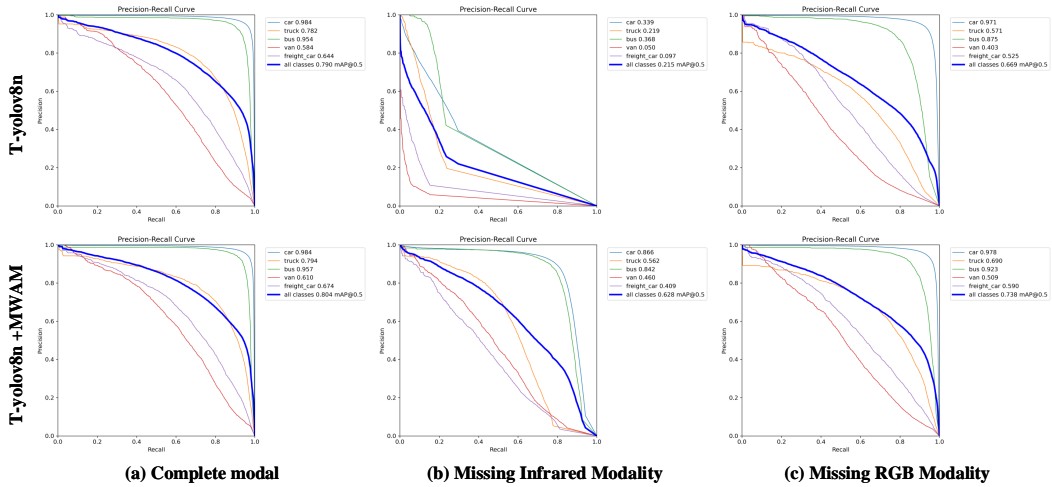

Figure 7: Comparative analysis of PR curves.

Table 14: Comparison with different balanced multimodal training methods. This table serves as a continuation of Table 13.

| Modal | | MSES | | MSLR | | MWAM | |
|---|---|---|---|---|---|---|---|
| Rgb | Depth | Acc(↑) | PCR(↓) | Acc | PCR | Acc | PCR |
| ✓ | ✗ | 85.86 | 8.57 | 70.72 | 25.96 | **93.99** | **4.96** |
| ✗ | ✓ | 84.95 | 9.54 | 79.79 | 16.47 | ***97.68*** | 1.22 |
| ✓ | ✓ | 93.91 | / | 95.52 | / | **98.89** | / |
| **Average** | | 88.24 | 9.06 | 82.01 | 21.22 | **96.85** | **3.09** |
| Rgb | Ir | Acc(↑) | PCR(↓) | Acc | PCR | Acc | PCR |
| ✓ | ✗ | 80.84 | 14.80 | 76.67 | 18.50 | **93.34** | ***3.40*** |
| ✗ | ✓ | 84.04 | 11.42 | 78.52 | 16.53 | **89.60** | **7.28** |
| ✓ | ✓ | 94.88 | / | 94.07 | / | **96.63** | / |
| **Average** | | 86.59 | 13.11 | 83.09 | 17.51 | **93.19** | **5.34** |
| Depth | Ir | Acc(↑) | PCR(↓) | Acc | PCR | Acc | PCR |
| ✓ | ✗ | 82.52 | 15.36 | 79.79 | 16.13 | **98.12** | **1.00** |
| ✗ | ✓ | ***79.67*** | ***18.28*** | 69.74 | 26.69 | **91.70** | **7.48** |
| ✓ | ✓ | 97.49 | / | 95.13 | / | **99.11** | / |
| **Average** | | 86.56 | 16.82 | 81.55 | 21.41 | **96.31** | **4.24** |

## A.14 LIMITATION

While our proposed MWAM has demonstrated consistent performance gains, we have identified several promising directions for future enhancement. Key directions for future research include mitigating the model's sensitivity to its initial training state. This is motivated by our observation that the proposed method performs better when trained from scratch than when initialized with pre-trained weights. We attribute this to the greater exploratory freedom in the gradient space afforded by random initialization. Conversely, a pre-trained model operates within a constrained, fine-tuning regime, which limits this exploration. Consequently, future work should focus on two areas: (1) enhancing the representational power of the pre-trained encoder, and (2) refining the formulation of quantitative evaluation metrics.

Furthermore, we plan to refine specific components of the model. The FRM module currently measures modality preference using frequency-based weights, but it does not explicitly disentangle the frequency contributions of each modality. A deeper analysis of these internal frequency dynamics could provide more granular control and potentially boost performance. Additionally, the fixed scaling factor in Eq. 5 represents a potential constraint. We will explore adaptive or learnable scaling

strategies to enhance the model's dynamic range and overall efficacy. These research avenues will form the core of our subsequent work.

## A.15 Comparison with More Multimodal Robustness Methods

Due to space constraints in the main paper, we present extended performance comparisons between MWAM and other state-of-the-art methods here. These evaluations cover experiments on both brain tumor segmentation and multimodal classification. The detailed results for the brain tumor segmentation and multimodal classification tasks are provided in Table 15 and Tables 16-17, respectively.

Table 15: Performance comparison of multimodal robust solutions on BRATS2020 dataset. This table serves as a continuation of Table 2.

| \multicolumn Modal | | | | HeMIS | | Robust Seg | | M3AE | | LS3M | | A2FSeg | |
|---|---|---|---|---|---|---|---|---|---|---|---|---|---|
| F | T1 | T1c | T2 | Dice(↑) | PCR(↓) | Dice | PCR | Dice | PCR | Dice | PCR | Dice | PCR |
| ✗ | ✗ | ✗ | ✓ | 79.85 | 6.27 | 82.20 | 8.13 | 85.17 | **4.99** | **86.34** | *6.26* | *85.36* | 6.52 |
| ✗ | ✗ | ✓ | ✗ | 64.58 | 24.19 | 71.39 | 20.21 | 77.65 | **13.38** | **79.36** | *13.84* | *77.86* | 14.73 |
| ✗ | ✓ | ✗ | ✗ | 63.01 | 26.04 | 71.41 | 20.19 | 74.88 | *16.47* | **79.35** | **13.85** | *75.86* | 16.92 |
| ✓ | ✗ | ✗ | ✗ | 52.29 | 38.62 | 82.87 | 7.38 | 85.86 | **4.22** | **88.14** | *4.30* | *86.46* | 5.31 |
| ✗ | ✗ | ✓ | ✓ | 84.45 | **0.87** | 85.97 | 3.91 | 86.51 | *3.49* | **88.01** | 4.45 | *87.74* | 3.91 |
| ✗ | ✓ | ✓ | ✗ | 72.50 | 14.90 | 76.84 | 14.12 | 78.18 | 12.78 | **82.73** | **10.18** | *80.59* | *11.74* |
| ✓ | ✓ | ✗ | ✗ | 65.29 | 23.36 | 88.10 | **1.53** | 87.98 | *1.85* | **88.93** | 3.45 | *88.85* | 2.69 |
| ✗ | ✓ | ✗ | ✓ | 82.31 | **3.38** | 85.53 | 4.40 | 86.34 | *3.68* | **87.30** | 5.22 | *87.87* | 3.77 |
| ✓ | ✗ | ✗ | ✓ | 81.56 | 4.26 | 88.09 | *1.54* | **89.20** | **0.49** | **90.48** | 1.76 | *89.20* | 2.31 |
| ✓ | ✗ | ✓ | ✗ | 69.37 | 18.57 | 87.33 | 2.39 | *88.36* | **1.43** | **90.43** | *1.82* | *88.36* | 3.23 |
| ✓ | ✓ | ✓ | ✗ | 73.31 | 13.95 | 88.87 | **0.67** | 88.46 | *1.32* | **90.29** | 1.98 | *89.32* | 2.18 |
| ✓ | ✓ | ✗ | ✓ | 83.03 | 2.54 | 89.24 | *0.26* | 89.54 | **0.11** | **90.79** | 1.43 | *89.96* | 1.48 |
| ✓ | ✗ | ✓ | ✓ | 84.64 | 0.65 | 88.68 | 0.88 | 89.54 | **0.11** | **90.11** | *2.17* | *90.00* | 1.43 |
| ✗ | ✓ | ✓ | ✓ | 85.19 | **0** | 86.63 | 3.17 | 86.94 | *3.01* | **88.27** | 4.16 | *87.64* | 4.02 |
| ✓ | ✓ | ✓ | ✓ | 85.19 | / | 89.47 | / | 89.64 | / | **92.11** | / | *91.31* | / |
| \multicolumn Average | | | | 75.10 | 12.68 | 84.17 | 6.34 | 85.62 | **4.81** | **87.51** | *5.35* | *86.43* | 5.73 |

Table 16: Performance comparison of multimodal robust solutions on SURF dataset. This table serves as a continuation of Table 4.

| \multicolumn Modal | | | SF-MD | | HeMIS | | RFNet | | MMANet | | SF-MD† | | MMANet† | |
|---|---|---|---|---|---|---|---|---|---|---|---|---|---|---|
| Rgb | Depth | Ir | Acc(↑) | PCR(↓) | Acc | PCR | Acc | PCR | Acc | PCR | Acc | PCR | Acc | PCR |
| ✓ | ✗ | ✗ | 83.92 | 13.83 | 85.64 | 12.64 | 87.57 | 11.38 | 91.43 | 7.77 | *92.13* | *6.87* | **93.49** | **5.32** |
| ✗ | ✓ | ✗ | 93.65 | 3.84 | 95.30 | 2.78 | 95.83 | 3.03 | *97.73* | *1.41* | 96.55 | 2.41 | **98.31** | **0.44** |
| ✗ | ✗ | ✓ | 89.60 | 8.00 | 83.79 | 14.53 | 85.31 | 13.67 | 89.96 | 9.25 | *93.76* | *5.23* | **94.18** | **4.62** |
| ✓ | ✓ | ✗ | 95.43 | 2.01 | 96.77 | 1.29 | 97.77 | 1.06 | *98.39* | 0.75 | 98.25 | *0.69* | **98.64** | **0.10** |
| ✓ | ✗ | ✓ | 93.26 | 4.24 | 93.73 | 4.39 | 95.73 | 3.13 | **96.99** | **2.16** | *96.62* | 2.33 | 96.60 | *2.17* |
| ✗ | ✓ | ✓ | 96.74 | 0.67 | 96.32 | 1.74 | 96.78 | 2.06 | *98.82* | 0.31 | 98.63 | 0.30 | **99.27** | **-0.54** |
| ✓ | ✓ | ✓ | 97.39 | / | 98.03 | / | 98.82 | / | **99.13** | / | *98.93* | / | 98.74 | / |
| \multicolumn Average | | | 92.85 | 5.43 | 92.82 | 6.23 | 93.98 | 5.72 | 96.06 | 3.61 | *96.41* | *2.97* | **97.03** | **2.02** |

Table 17: Comparison with other methods for handling missing modalities. To ensure a fair comparison, we only report results on the SURF dataset using the two most dissimilar modalities: RGB and Depth.

| \multicolumn Modal | | CRMT-JT | | COM | | DCP | | MWAM | |
|---|---|---|---|---|---|---|---|---|---|
| Rgb | Depth | Acc(↑) | PCR(↓) | Acc | PCR | Acc | PCR | Acc | PCR |
| ✓ | ✗ | 93.08 | 5.42 | 92.79 | 5.28 | 93.01 | 5.74 | **93.99** | **4.96** |
| ✗ | ✓ | 96.74 | 1.70 | 96.82 | *1.16* | **98.15** | **0.53** | *97.68* | 1.22 |
| ✓ | ✓ | 98.41 | / | 97.96 | / | *98.67* | / | **98.89** | / |
| \multicolumn Average | | 96.08 | 3.56 | 95.86 | 3.22 | *96.61* | *3.13* | **96.85** | **3.09** |

## A.16   Performance on Fine-Grained Classification Tasks

Given that the design of FRM prioritizes the role of low-frequency components across modalities, a natural question arises: Is the effectiveness of our MWAM limited to low-frequency-dominant tasks, and does it underperform in high-frequency-dominant scenarios? To investigate this concern, we conduct experiments on fine-grained classification, a task inherently driven by high-frequency details, to demonstrate the broad applicability and robustness of MWAM.

### A.16.1   The Increased Reliance of Fine-Grained Classification on High-Frequency Components

We first conduct a series of experiments to demonstrate that fine-grained classification is a high-frequency dominant task. As existing public datasets for fine-grained classification do not meet our specific experimental requirements, we begin by constructing a custom dataset through the following procedure: We select the classic FGVC Aircraft dataset (30 classes) and then utilize the DepthAnything v2 model to generate a corresponding depth map for each RGB image, creating a paired dataset.

For our experimental setup, we employ a vanilla ResNet18 as the backbone classifier. We evaluate performance based on classification Accuracy and the PCR across various modality combinations. A total of 14 experimental sets were conducted. In seven sets, the dataset was processed using low-pass filters, while in the remaining seven, high-pass filters with identical kernel sizes were applied. We investigate two ranges of kernel sizes: small kernels ($3\times3$, $5\times5$, $7\times7$, and $9\times9$) and large kernels ($17\times17$, $31\times31$, and $51\times51$).

Visualizations of the filtered data are provided in Figure 8. The comparative experimental results are summarized in Tables 18 to 21, and the performance curves are plotted in Figure 9.

The experimental results reveal several intriguing phenomena. First, for the low-pass filter group, performance exhibits a rise-then-fall pattern as the kernel size increases. We posit that with small kernels, the low-pass filter effectively removes imperceptible high-frequency artifacts, such as Gaussian noise. Given the $518\times518$ image resolution, these small kernels are insufficient to smooth out task-relevant high-frequency details like edges. This noise removal enhances the robustness of the model's predictions, leading to a slight performance gain. Subsequently, as the kernel size grows, the filter begins to erode these critical high-frequency features. This detrimental effect becomes more pronounced with larger kernels, causing a sharp degradation in performance.

In stark contrast, the control group using high-pass filters demonstrates a much more gradual decline in performance, even as the kernel size increases. Notably, it avoids the catastrophic collapse in classification accuracy observed in the low-pass scenario with large kernels.

From these observations, we can deduce that high-frequency components play a dominant role in fine-grained classification, far more so than in general classification tasks. Furthermore, these critical signals are not only highly sensitive but are also intricately intertwined with complex high-frequency noise, which constitutes a core challenge in fine-grained recognition.

### A.16.2   Comparative Experiments on Fine-grained Classification

To verify the versatility of our approach, we conducted a fine-grained classification task. Specifically, we selected the Stanford Dogs dataset (120 classes) and the FGVC Aircraft dataset (30 classes), generating corresponding depth representations using DepthAnything v2. We chose the basic ResNet18 structure as the baseline to verify the classification accuracy before and after adding MWAM. The experimental results are presented in Table 22.

The results show that MWAM consistently boosts ResNet18's performance in fine-grained classification tasks. This indicates our method works well not only for low-frequency tasks but also for high-frequency ones.

Furthermore, Figure 10 compares the overall performance of the Baseline model against our Baseline+MWAM under various filtering conditions. The results highlight four key phenomena:

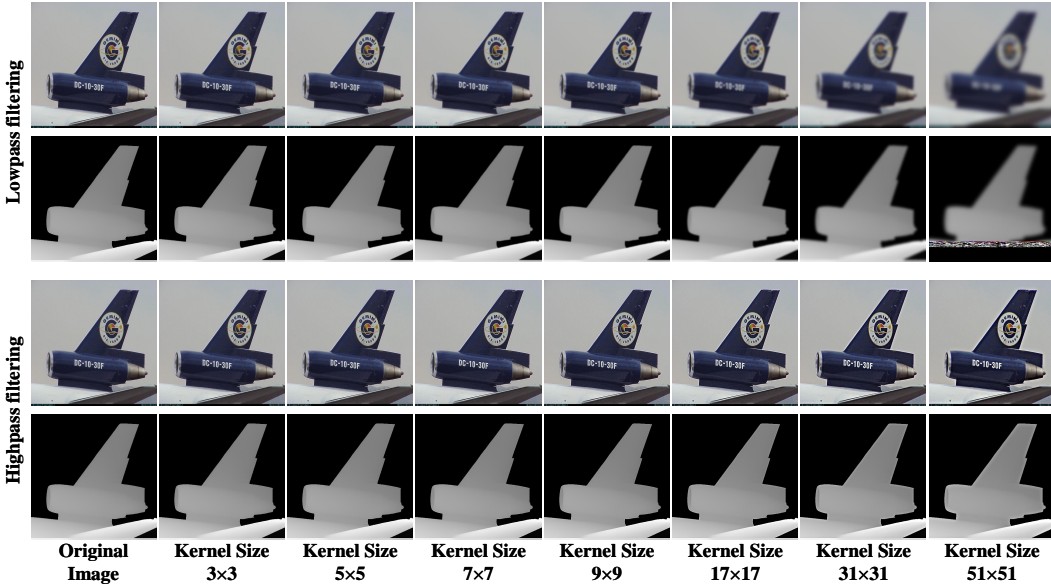

Figure 8: Visualization of filtering effects on dataset samples. This figure showcases the comparative results of applying low-pass and high-pass filters with various kernel sizes to paired visible-light and depth images.

Table 18: Performance comparison of low-pass filters with varying small kernel sizes.

| Modal | | w/o filter | | 3×3 | | 5×5 | | 7×7 | | 9×9 | |
|---|---|---|---|---|---|---|---|---|---|---|---|
| RGB | Depth | Acc(↑) | PCR(↓) | Acc | PCR | Acc | PCR | Acc | PCR | Acc | PCR |
| ✓ | ✗ | 61.60 | 9.57 | 66.95 | 6.53 | 67.22 | 7.03 | 62.77 | 11.03 | 56.73 | 15.98 |
| ✗ | ✓ | 39.81 | 41.56 | 39.54 | 44.80 | 40.02 | 44.65 | 39.78 | 43.61 | 38.58 | 42.86 |
| ✓ | ✓ | 68.12 | / | 71.63 | / | 72.30 | / | 70.55 | / | 67.52 | / |
| **Average** | | 56.51 | 25.57 | 59.37 | 25.67 | 59.85 | 25.84 | 57.70 | 27.32 | 54.28 | 29.42 |

**Effectiveness of MWAM:** Augmenting the Baseline with MWAM yields a substantial and consistent performance improvement across nearly all filter settings (with the exception of the 51x51 low-pass filter). This primarily demonstrates the effectiveness of our proposed module.

**Impact of High-Frequency Noise:** Both the Baseline and Baseline+MWAM exhibit similar performance trends under low-pass and high-pass filtering. This suggests the presence of subtle, high-frequency noise in the dataset that, while imperceptible to humans, adversely affects the model's training process.

**Attention to High-Frequency Information:** When using large-kernel low-pass filters, the performance of Baseline+MWAM is significantly lower than its counterpart under high-pass filtering with the same kernel size. This indicates that for tasks dominated by high-frequency features, MWAM effectively utilizes the high-frequency information in the images.

**Task Degradation under Extreme Smoothing:** When the kernel size reaches 51x51, the excessive smoothing from the low-pass filter results in comparable performance between the two models (with Baseline+MWAM even slightly underperforming). This implies that under such conditions, the model may degrade the fine-grained task into a simpler one, such as color recognition, which relies heavily on low-frequency components.

In summary, these findings demonstrate the versatility of MWAM: it not only excels at conventional low-frequency-dominant image understanding tasks but is also highly effective for tasks that depend on high-frequency information.

Table 19: Performance comparison of low-pass filters with varying large kernel sizes.

| Modal | | w/o filter | | 17×17 | | 31×31 | | 51×51 | |
|---|---|---|---|---|---|---|---|---|---|
| RGB | Depth | Acc(↑) | PCR(↓) | Acc | PCR | Acc | PCR | Acc | PCR |
| ✓ | ✗ | 61.60 | 9.57 | 31.70 | 28.76 | 18.33 | 19.11 | 14.75 | 4.10 |
| ✗ | ✓ | 39.81 | 41.56 | 31.55 | 29.10 | 21.72 | 4.15 | 16.92 | -10.01 |
| ✓ | ✓ | 68.12 | / | 44.50 | / | 22.66 | / | 15.38 | / |
| **Average** | | 56.51 | 25.57 | 35.92 | 28.93 | 20.90 | 11.63 | 15.68 | -2.96 |

Table 20: Performance comparison of high-pass filters with varying small kernel sizes.

| Modal | | w/o filter | | 3×3 | | 5×5 | | 7×7 | | 9×9 | |
|---|---|---|---|---|---|---|---|---|---|---|---|
| RGB | Depth | Acc(↑) | PCR(↓) | Acc | PCR | Acc | PCR | Acc | PCR | Acc | PCR |
| ✓ | ✗ | 61.60 | 9.57 | 56.49 | 12.35 | 54.51 | 13.32 | 52.22 | 14.10 | 50.21 | 14.57 |
| ✗ | ✓ | 39.81 | 41.56 | 39.69 | 38.42 | 39.66 | 36.94 | 39.60 | 34.86 | 39.42 | 32.92 |
| ✓ | ✓ | 68.12 | / | 64.45 | / | 62.89 | / | 60.79 | / | 58.77 | / |
| **Average** | | 56.51 | 25.57 | 53.54 | 25.38 | 52.35 | 25.13 | 50.87 | 24.48 | 49.47 | 23.75 |

Table 21: Performance comparison of high-pass filters with varying large kernel sizes.

| Modal | | w/o filter | | 17×17 | | 31×31 | | 51×51 | |
|---|---|---|---|---|---|---|---|---|---|
| RGB | Depth | Acc(↑) | PCR(↓) | Acc | PCR | Acc | PCR | Acc | PCR |
| ✓ | ✗ | 61.60 | 9.57 | 46.51 | 16.11 | 44.62 | 13.76 | 43.75 | 9.40 |
| ✗ | ✓ | 39.81 | 41.56 | 37.98 | 31.49 | 33.23 | 35.78 | 29.48 | 38.95 |
| ✓ | ✓ | 68.12 | / | 55.44 | / | 51.74 | / | 48.29 | / |
| **Average** | | 56.51 | 25.57 | 46.64 | 23.80 | 43.20 | 24.77 | 40.51 | 24.18 |

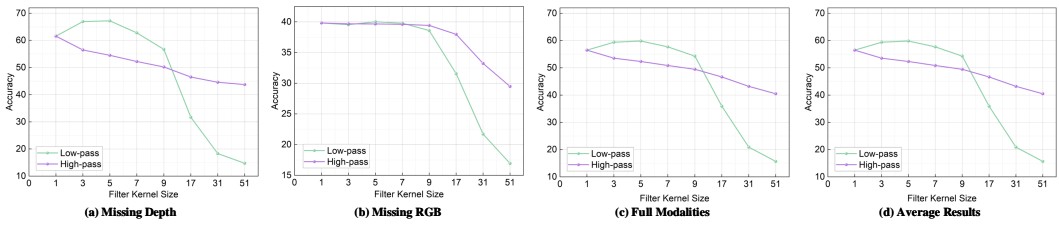

Figure 9: Effect of filter type and kernel size on model performance.

Table 22: Validation of the effectiveness of MWAM on fine-grained classification tasks. We selected the Stanford Dogs dataset and the FGVC Aircraft dataset, and used the DepthAnything v2 model to generate corresponding depth representations. The classification performance before and after deploying our MWAM is tested using the vanilla ResNet18 structure as the base model.

| Modal | | Stanford Dogs Dataset | | | | FGVC-Aircraft Dataset | | | |
|---|---|---|---|---|---|---|---|---|---|
| | | w/o MWAM | | w/ MWAM | | w/o MWAM | | w/ MWAM | |
| Rgb | Depth | Acc(↑) | PCR(↓) | Acc | PCR | Acc | PCR | Acc | PCR |
| ✓ | ✗ | 32.91 | 38.73 | **43.91** | **28.84** | 61.60 | 9.57 | **74.85** | **6.21** |
| ✗ | ✓ | 27.90 | 48.05 | **36.20** | **41.34** | 39.81 | 41.56 | **56.22** | **29.56** |
| ✓ | ✓ | 53.71 | / | **61.71** | / | 68.12 | / | **79.81** | / |
| **Average** | | 38.17 | 43.39 | **47.27** | **35.09** | 56.51 | 25.57 | **70.29** | **17.89** |

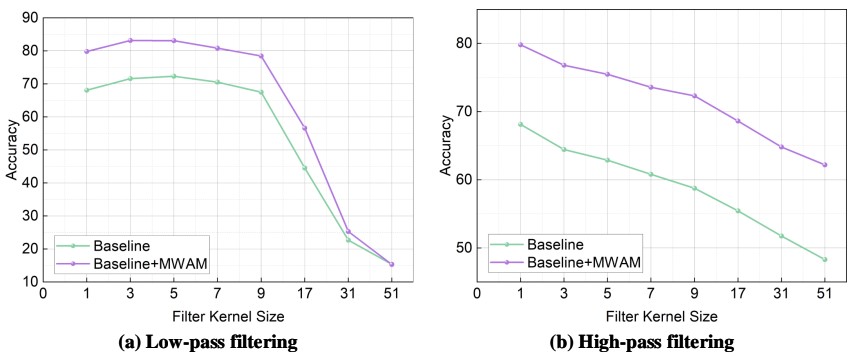

Figure 10: Performance comparison of methods under different filters.

