# OpenReview forum: "Plug, Play, and Fortify: A Low-Cost Module for Robust Multimodal Image Understanding Models"
_ICLR.cc/2026/Conference — ICLR 2026 Poster_

### Official Review · Reviewer_K6MD · 2025-10-22

**Soundness:** 2
**Presentation:** 2
**Contribution:** 2
**Rating:** 2
**Confidence:** 4

**Summary:**

This paper presents a novel approach to multimodal learning that applies the Discrete Cosine Transform (DCT).
The purpose of using DCT is to extract the low- and high-frequency components of each modality in the frequency domain, thereby capturing the intrinsic characteristics of each modality.
This approach is employed in two key modules:

- **Frequency Ratio Metric (FRM)**: Measures the dominance of each modality.
- **Multimodal Weight Allocation Module (MWAM)**: Dynamically allocates modality weights within each batch to determine the relative contribution of each modality.

This design leads to improved performance in several experimental settings.

**Strengths:**

- **Strength 1**: Preprocessing input data in the frequency domain can enhance efficiency in capturing the power and meaningful information of each modality within the input space.
- **Strength 2**: Since this approach focuses on extracting informative representations from the input space, it can be broadly and effectively applied to various vision tasks, such as segmentation, classification, and object detection.

**Weaknesses:**

- **Weakness 1**: Although this paper focuses on multimodal learning, its proposed approach is primarily applicable to vision-based tasks. It might seems difficult to extend the method to other modalities, such as text, where applying DCT directly is not feasible. Additional processing steps would be required to make DCT applicable to such modalities.
- **Weakness 2**: The method involves a large number of hyperparameters that can significantly influence the results. For example, variations in $\alpha$, $\beta$, $\lambda$, $\gamma$, and $\sigma$ in Eq.(4) and (5) may lead to substantially different outcomes. This complexity makes it challenging to determine which hyperparameters are critical or not. Although the authors provide ablation studies in Table 9, a clearer justification of the importance of each hyperparameter would strengthen the work. Otherwise, it would be better to change the reflected sigmoid function used in the weight allocation process.
- **Weakness 3**: The performance analysis lacks clarity in certain aspects. For instance, in Table 2, results for the vanilla GSS model are missing, even though GSS + MWAM achieved the best overall performance. The authors compare this with LS3M; however, since GSS and LS3M are fundamentally different approaches, this comparison may not be entirely fair. It would be more appropriate to compare Original vs. Original + MWAM to more fairly and effectively highlight the contribution and novelty of MWAM. Same comments for other Tables.
- **Weakness 4**: The overall structure of the paper and writings could be improved for better coherence. For instance, in Section 3.1, the authors present empirical analyses on the influence of different frequency components, while all theoretical discussions are deferred to the appendix. Given that one of the paper’s novelty is the theoretical and empirical validation of modality dominance, I think it is required to include at least a concise summary or key results of the theoretical validations within the main text, even if the detailed derivations remain in the appendix.
Additionally, I would recommend that important equations and their explanations be formatted like Definitions, Lemmas, and Theorems. (e.g., Appendix A.1)


Minor Issues:

Words change at Abs \& Conclusion: **efficiency** method $\Rightarrow$ **efficient** method

Typo at line 296: show the sensitivity **o** these factors  $\Rightarrow$ show the sensitivity **of** these factors

**Questions:**

- **Question 1**: What is the motivation for applying the Discrete Cosine Transform (DCT) instead of other frequency-domain transforms such as the Discrete Fourier Transform (DFT) or Fast Fourier Transform (FFT)? In vision tasks, DFT/FFT captures both magnitude and phase information, whereas DCT focuses on real-valued frequency components and is often used for compression. It would be helpful to understand why DCT was specifically chosen in this context and how it benefits the representation of visual modalities.
- **Question 2**: What are the main differences between this paper and other frequency-domain deep learning approaches? Previous studies have already applied Fourier transforms either between layers [1] or at the input space [2]. From the paper, the primary difference seems to be the use of DCT instead of FFT. Since both operate in the frequency domain, this substitution alone does not convincingly demonstrate methodological novelty.
- **Question 3**: What is the rationale for using the reflected sigmoid function with multiple hyperparameters in the Eq. (4)? This design choice seems critical, as it raises concerns that the observed performance improvements might depend heavily on hyperparameter tuning rather than the proposed MWAM mechanism itself. I could not find a clear explanation or reference supporting this design choice, and those 4 hyperparameters.
- **Question 4**: Have the authors considered evaluating MWAM on datasets that include additional modalities such as audio or text? This would help demonstrate the generalizability of the proposed approach beyond purely visual modalities.

[1] Guo, Shi, et al. "Spatial-frequency attention for image denoising." (preprint)

[2] He, Xilin, et al. "Towards combating frequency simplicity-biased learning for domain generalization." (NeurIPS 2024)

---

> ### Author Response · Authors · 2025-11-13
> **Response to Reviewer K6MD's Comments (1/5)**
>
> Dear Reviewer **K6MD**:
>
> Thank you for your thorough review and insightful feedback. We are pleased that you found our methodology and experimental thoroughness to be strengths of the work.
>
> Based on your comments, we have identified a fundamental disconnect between your interpretation and our intended contributions across several areas, including the work's motivation, methodology, and experimental design. We believe these points of divergence may have made it difficult to fully assess the novelty and significance of our work. Therefore, we would like to offer detailed clarifications on these specific issues, which we hope will enable a more informed re-examination of our manuscript.
>
> Next, I will help clarify the misunderstanding.
>
> **I. Summary**
>
> We focus on improving the robustness of incomplete **multi-modal visual understanding models**. We attribute the performance collapse caused by modal absence to the incomplete optimization of the internal branches of the model due to modal competition, and attempt to alleviate this collapse phenomenon by balancing the model's attention to different modalities. First, through experiments and theoretical derivations, we discover and prove that in multi-modal visual classification tasks, the low-frequency components in images will, to a certain extent, dominate the model training. Subsequently, we design FRM and MWAM to evaluate the model's preference for different modalities and adopt positive treatment methods.
>
> **II. Comments and Responses**
>
> **W1:** Although this paper focuses on multimodal learning, its proposed approach is primarily applicable to vision-based tasks. It might seems difficult to extend the method to other modalities, such as text, where applying DCT directly is not feasible. Additional processing steps would be required to make DCT applicable to such modalities.
>
> **R1:** Thank you for your comment. We agree that our method is tailored for **vision-based modalities**. This was a deliberate design choice, and it forms the core of our contribution.
>
> Our paper's scope is explicitly defined as **multi-modal image understanding**, a point we stress **in the title and throughout the manuscript**. Therefore, the applicability to **vision-based tasks** is not a secondary outcome but the central focus of our investigation.
>
> While we clearly define this scope, we also recognize its boundaries. As mentioned in our "Limitations" section, extending this work to more diverse modalities is a primary goal for our future research. We are pleased to report that **we are already well underway with a more generalizable solution**. Our next-generation framework moves beyond the input data space—the main constraint of the current method—and operates in the latent feature space. By analyzing the dynamics of latent representations, we can address modality imbalance in a manner that is fundamentally agnostic to the initial data encoding, making it suitable for a much broader set of modalities.
>
> **W2:** The method involves a large number of hyperparameters that can significantly influence the results. For example, variations in $\alpha$, $\beta$, $\lambda$, $\gamma$, and $\sigma$ in Eq.(4) and (5) may lead to substantially different outcomes. This complexity makes it challenging to determine which hyperparameters are critical or not. Although the authors provide ablation studies in Table 9, a clearer justification of the importance of each hyperparameter would strengthen the work. Otherwise, it would be better to change the reflected sigmoid function used in the weight allocation process.
>
> **R2:** Thank you for your comment. Indeed, all four of the aforementioned scaling factors serve a single, unified objective: to calibrate the output weight based on the numerical disparities in T (as defined in Eq. 5). This objective is achieved by leveraging these four factors to precisely modulate the shape of the Sigmoid non-linear function.
>
> As described in our manuscript, the relevant explanations can be found on **lines 292-296** and are further detailed in **lines 1134-1174**. For a more comprehensive discussion on this matter, we would like to refer you to our **Answer to Question 3**.

---

> ### Author Response · Authors · 2025-11-13
> **Response to Reviewer K6MD's Comments (2/5)**
>
> **W3:** The performance analysis lacks clarity in certain aspects. For instance, in Table 2, results for the vanilla GSS model are missing, even though GSS + MWAM achieved the best overall performance. The authors compare this with LS3M; however, since GSS and LS3M are fundamentally different approaches, this comparison may not be entirely fair. It would be more appropriate to compare Original vs. Original + MWAM to more fairly and effectively highlight the contribution and novelty of MWAM. Same comments for other Tables.
>
> **R3:** Thank you for your comment. We would like to clarify that the results for the **vanilla GSS are not missing**, but are presented in **Table 16 (Line 1407)** in the appendix. As indicated in the captions of both **Table 2** and **Table 16**, we established this link to guide the reader.
>
> Due to the strict page limits of the main paper, we were unable to present all detailed comparisons side-by-side. Therefore, we adopted the approach of using tables in the main text to highlight key findings and linking them to comprehensive result tables in the appendix.
>
> The primary intention of comparing GSS and LS3M in Table 2 was **to demonstrate that by incorporating our MWAM module, an earlier method (GSS) can achieve performance competitive with a state-of-the-art (SOTA) method (LS3M).** Our goal was not to make a direct architectural comparison between these two methods, a point we discuss in the manuscript **(Lines 375-394)**.
>
> By comparing Table 2 (GSS+MWAM) and Table 16 (vanilla GSS), one can see that GSS+MWAM clearly outperforms the vanilla GSS baseline, confirming the effectiveness of our module.
>
> **W4:** The overall structure of the paper and writings could be improved for better coherence. For instance, in Section 3.1, the authors present empirical analyses on the influence of different frequency components, while all theoretical discussions are deferred to the appendix. Given that one of the paper’s novelty is the theoretical and empirical validation of modality dominance, I think it is required to include at least a concise summary or key results of the theoretical validations within the main text, even if the detailed derivations remain in the appendix. Additionally, I would recommend that important equations and their explanations be formatted like Definitions, Lemmas, and Theorems. (e.g., Appendix A.1)
>
> **R4:** Thank you for this valuable suggestion. We will carefully consider your point and make appropriate revisions to enhance clarity without disrupting the overall narrative flow of the paper.
>
> We would like to clarify the rationale for our current structure. The manuscript is organized along a central storyline: 'Observation → Analysis → Problem Formulation → Proposed Solution → Experimental Validation.' This top-down approach is designed to guide the reader through our core logic and contributions seamlessly. To maintain this focused narrative, we intentionally placed the detailed mathematical derivations in the appendix, as they primarily serve as supplementary proof rather than being central to the main argument. We believe this organization best serves the reader by keeping the main text concise and accessible.

---

> ### Author Response · Authors · 2025-11-13
> **Response to Reviewer K6MD's Comments (3/5)**
>
> **Q1:** What is the motivation for applying the Discrete Cosine Transform (DCT) instead of other frequency-domain transforms such as the Discrete Fourier Transform (DFT) or Fast Fourier Transform (FFT)? In vision tasks, DFT/FFT captures both magnitude and phase information, whereas DCT focuses on real-valued frequency components and is often used for compression. It would be helpful to understand why DCT was specifically chosen in this context and how it benefits the representation of visual modalities.
>
> **A1:** Thank you for your thoughtful comment regarding our choice of frequency transformation technique. This is an excellent point, and we appreciate the opportunity to clarify our rationale.
>
> The primary and sole purpose of employing frequency analysis in our framework is to **calculate the FRM**. As defined in our manuscript, the FRM calculation is based exclusively on the energy distribution between low-frequency and high-frequency components of the input modalities; it **does not require phase information**.
>
> You are correct that several tools, such as FFT, DFT, and DCT, can perform frequency transformation. However, they possess distinct properties that make them more or less suitable for specific tasks. Our decision to use DCT was driven by **two key advantages** it offers for our particular application:
>
>    1. **Superior Energy Compaction:** DCT is renowned for its excellent energy compaction properties, especially for highly correlated data like images. It efficiently concentrates a large portion of the signal's energy into a few low-frequency coefficients.
>
>    2. **Intuitive Frequency Partitioning:** The 2D-DCT has an inherent and highly convenient spatial layout where low-frequency components are aggregated in the top-left corner and high-frequency components in the bottom-right. This structure allows us to directly and easily partition the frequency bands and compute their respective energies to derive the FRM.
>
> While we do not dispute that FFT-based methods could be adapted to serve the same function, they would be less direct. Using FFT would necessitate additional design choices and processing steps, such as handling complex-valued outputs and implementing a specific logic to separate and quantify energy from the centered frequency spectrum.
>
> In summary, DCT provides a more direct, elegant, and computationally straightforward solution for the specific task of calculating the FRM, which is the cornerstone of our MWAM framework.
>
> **Q2:** What are the main differences between this paper and other frequency-domain deep learning approaches? Previous studies have already applied Fourier transforms either between layers [1] or at the input space [2]. From the paper, the primary difference seems to be the use of DCT instead of FFT. Since both operate in the frequency domain, this substitution alone does not convincingly demonstrate methodological novelty.
>
> **A2:** Thank you for your comment. MWAM is not equivalent to [1] or [2]. The distinctions are fundamental.
>
> **MWAM vs. [1]:**
>
> 1. **Task:** [1] is for low-level image denoising. We are committed to enhancing the robustness of multimodal image understanding models under modal missing conditions.
>
> 2. **Motivation:** [1] uses frequency to improve attention efficiency. We use it to diagnose and fix modality imbalance.
>
> 3. **Method:** [1] modifies features within a network pipeline. We analyze input modalities to dynamically assign training weights. The only commonality is the use of a frequency tool, not the idea.
>
> **MWAM vs. [2]:** We have cited [2] in our manuscript (line 139) and appreciate its contribution. However, it is distinct from our work:
>
> 1. **Task:** [2] tackles single-modality domain generalization by fighting "frequency shortcuts." We tackle robustness to missing modalities in a multi-modal setting. These are separate fields.
>
> 2. **Method:** [2] alters dataset statistics in the frequency domain. MWAM calculates a metric (FRM) from input modalities and uses it to re-weight their influence during training (via gradients or loss). The frameworks are entirely different.
>
> In summary, while these references also use frequency analysis, **they do so to solve completely different problems with different methods**. Their connection to our work is superficial and does not challenge the novelty of MWAM, which introduces a new approach specifically for the problem of modality imbalance in multi-modal image understanding systems.
>
> [1] Guo, Shi, et al. "Spatial-frequency attention for image denoising." (preprint)
>
> [2] He, Xilin, et al. "Towards combating frequency simplicity-biased learning for domain generalization." (NeurIPS 2024)

---

> > ### Comment · Reviewer_K6MD · 2025-11-13
> > **Responses to the comments (3,4,5/5)**
> >
> > **A1**:
> > Thank you for the clarification. I was merely interested in understanding the rationale behind the authors’ decision to employ the Discrete Cosine Transform (DCT) instead of the Fourier Transform (FT).
> >
> > **A2**:
> > I appreciate the authors’ detailed response and apologize for any earlier ambiguity in my question. My intention was to highlight that, from a fundamental standpoint, the underlying concept of training within the frequency (sub-)space or domain remains conceptually similar, even if the specific objectives and technical implementations differ. From my perspective, the proposed approach appears somewhat incremental in terms of novelty. Nevertheless, I acknowledge that I may be overly cautious in this interpretation, and I would welcome a deeper discussion on this point.
> >
> > **A3**: Discussed in R2
> >
> > **A4**:
> > I apologize once again for any confusion or misunderstanding. My concerns regarding this point have been fully resolved.
> >
> > &nbsp;
> >
> > ==============================================
> >
> > I would be very happy to engage in a deeper and further discussion regarding the points raised in R1-4 and A1-4.

---

> > > ### Author Response · Authors · 2025-11-13
> > > **Response to the following comments of Reviewer K6MD according (3,4,5/5)**
> > >
> > > Hi!
> > >
> > > Perhaps the extensive details in our manuscript inadvertently obscured our core message, and for any lack of clarity, we sincerely apologize. We are grateful for the opportunity to address your concerns and to engage in this deeper discussion about the motivation behind our research.
> > >
> > > You are correct in observing that both MWAM and the cited works leverage the power of frequency-domain analysis. This can be attributed to the fact that it is a powerful analytical paradigm, offering a more global perspective (e.g., via energy aggregation) to address fundamental problems in image analysis.
> > >
> > > However, we believe a crucial distinction must be made, which lies at the heart of our contribution: **the novelty of our work is not merely in using a frequency-domain method to solve a problem, but in adopting a frequency-domain perspective to analyze and understand the problem itself.**
> > >
> > > To be specific, our foundational discovery was that the imbalance and competition between modalities can be effectively characterized and quantified in the frequency domain. From this viewpoint, **DCT is simply an instrumental choice**—a means to an end—for extracting features to compute our FRM metric.
> > >
> > > **More important than the specific method**, we argue, is the act of opening up the study of modality missingness and competition through this frequency lens. This is a critical viewpoint that has been largely overlooked by existing work in the field. It was this key insight that motivated the development of MWAM as a framework to balance model training and ultimately enhance robustness.
> > >
> > > Thank you once again for your time and insightful feedback. We look forward to your further reply, and wish you a pleasant day.

---

> ### Author Response · Authors · 2025-11-13
> **Response to Reviewer K6MD's Comments (4/5)**
>
> **Q3:** What is the rationale for using the reflected sigmoid function with multiple hyperparameters in the Eq. (4)? This design choice seems critical, as it raises concerns that the observed performance improvements might depend heavily on hyperparameter tuning rather than the proposed MWAM mechanism itself. I could not find a clear explanation or reference supporting this design choice, and those 4 hyperparameters.
>
> **A3:** Thank you for your observation. You are correct to point out the sensitivity of the parameters in Equations 4 and 5; this is, in fact, by design.
>
> This mechanism is a critical component of MWAM, requiring meticulous tuning. **The four hyperparameters govern the translation and scaling of the sigmoid function along both axes, granting it the adaptability needed**. For this reason, all four are essential, even if their value is 1 in some configurations.
>
> We discuss the motivation for this design on **lines 292-296** and provide further experimental details on **lines 1134-1174**.
>
> Based on the experimental results shown in **Table 9**, we concluded that the model is not highly sensitive to these parameters, provided they are chosen within a reasonable range.
>
> As noted in the manuscript, the role of these scaling factors is **to adjust the shape of the standard Sigmoid function**. This modification enables the resulting non-linear function to allocate appropriate weights to different modalities. Here, we would like to elaborate on our empirical tuning procedure for these factors.
>
> First, to provide some context, **our goal is to adjust the shape of a Sigmoid function where T (from Equation 5) is the independent variable**. The adjusted function must satisfy the following three requirements:
>
> 1. **Monotonicity:** The function must be monotonically decreasing, meaning a larger Feature Regulation Module (FRM) score (indicating dominance) results in a smaller weight.
> 2. **Effective Weight Allocation:** The function must assign meaningful and distinct weights based on the T values of different modalities. These weights should not all be less than 1, as this would suppress the overall training process (as detailed in Lines 1160-1163). Furthermore, since the T values are normalized and their relative differences can be small, the function needs to be sufficiently sensitive to these subtle differences to clearly reflect the model's preference for each modality, preventing the allocated weights from being influenced by noise.
> 3. **Non-Saturation:** The operating points (i.e., the T values) should fall within the approximately linear region of the Sigmoid curve, not its saturation zones, to ensure a responsive adjustment.
>
> With these requirements in mind, we employ the following two-step tuning methodology: We set the value '1' as the threshold separating dominant and weaker modalities, where dominant modalities receive weights less than 1 and weaker modalities receive weights greater than 1. These weights are then non-linearly scaled according to the differences in their respective T values.
>
> The tuning process is as follows:
>
> 1. **Initial Analysis:** We first statistically analyze the T values for each modality over a number of initial training iterations (the number depends on the dataset size). This provides a preliminary understanding of the typical numerical gap between the modalities' T values for the given task.
>
> 2. **Shape Adjustment:** We then use the four scaling factors to adjust the Sigmoid function's shape. We observe the resulting weights assigned to the modalities and tune the factors until the three requirements listed above are met.
>
> The necessity for these hyperparameters arises because different modality combinations yield different ranges and gaps in their calculated T values. The scaling factors provide the flexibility to adapt the function's output response to satisfy requirements 2 and 3 for any given scenario.
>
> In summary, by following this two-step empirical tuning method, we can effectively determine appropriate values for the scaling factors. Ultimately, all four factors serve a single, unified purpose: to reshape the standard Sigmoid function into a customized curve that is optimally suited for our modality weight allocation strategy.

---

> ### Author Response · Authors · 2025-11-13
> **Response to Reviewer K6MD's Comments (5/5)**
>
> **Q4:** Have the authors considered evaluating MWAM on datasets that include additional modalities such as audio or text? This would help demonstrate the generalizability of the proposed approach beyond purely visual modalities.
>
> **A4:** Thank you for this insightful comment regarding the generalizability of our method. We would like to clarify that the core focus of our research is on **multimodal image understanding models**, a scope we have emphasized in the title and throughout the manuscript.
>
> In fact, we did conduct preliminary explorations to test the feasibility of applying MWAM to non-image modalities, but the results were suboptimal. We found that the operational mechanism of MWAM is intrinsically designed for the characteristics of the image domain, which limits its direct applicability to other types of data. Based on these findings, we deliberately refined the manuscript's title and content to concentrate on solving multimodal image understanding tasks, such as medical image segmentation and semantic segmentation.
>
> That being said, we fully acknowledge this limitation. In the 'Limitations' section, we discuss the need for developing balancing strategies for a more diverse set of modalities beyond image-centric tasks. We have designated this as a key direction for our future work.
>
> To be more specific, **we have already commenced this follow-up research**. Our new work investigates how to balance a wider array of modality combinations by leveraging frequency domain analysis. We hypothesize that the primary constraint of MWAM stems from its operational domain—the input data space. To overcome this, our subsequent work transitions from the input space to a latent space perspective, conceptualized through frequency domain processing. In this new paradigm, we aim to define and address inter-modality imbalance by evaluating the dynamic changes in the latent features of different modalities, rather than acting on the raw input data.
>
> Once again, we thank you for your insightful comments. Should any part of our work or responses require further clarification, we would be happy to discuss it further with you during the rebuttal period.
>
> **Best regards,**
>
> authors of #2549.

---

> ### Comment · Reviewer_K6MD · 2025-11-13
> **Responses to the comments (1,2/5)**
>
> Thank you for the response. Some of my concerns have been addressed; however, a few issues still remain.
>
> **R1**: Thank you for the clarification. I apologize for the earlier misunderstanding. After completing my review, I realized that the authors have indeed stated that this paper focuses on vision-based multimodal learning. I appreciate the clarification and apologize for the confusion.
>
> **R2**: I definitely have read lines 292–296 and 1134–1174 during review. However, I remain concerned that the model’s performance may depend heavily on the choice of hyperparameters $\alpha$, $\beta$, $\gamma$, and $\sigma$. In my view, the authors should provide a clearer justification for why this sigmoid-based function is necessary. According to the response (comment 4/5), these hyperparameters are used to adjust the shape of the standard sigmoid function. However, since the authors have fixed these values for scaling purposes, this design decision may significantly influence the results. While this may appear to be a minor issue, if the performance is highly sensitive to these parameters, it could potentially undermine the validity of the approach and limit the main novelty of the paper.
>
> As a simple suggestion, similar to the temperature parameter in the InfoNCE loss, it might be beneficial to consider making these hyperparameters learnable rather than fixed.
>
> I understand that defining a specific range for these hyperparameters might be necessary, and I acknowledge that my concern could appear somewhat strict. Nevertheless, from my perspective, the core value of research lies in designing methods that are robust and adaptable across diverse tasks and datasets. Therefore, in my view, if the model’s performance depends too strongly on these hyperparameter settings—such as through loss scaling—the contribution may be less generalizable and its practical impact more limited.
>
> **R3 \& R4**
> Thank you for the clarification, and I apologize for overlooking some of the results (e.g., Table 16). However, I would still strongly recommend reorganizing the performance results to improve clarity and readability. Additionally, regarding my earlier comment (W4), my suggestion was that if the proposed mathematical approaches are critical to the paper’s main contribution, it would be preferable to present their key points in the main section rather than placing all of them in the appendix.

---

> > ### Author Response · Authors · 2025-11-13
> > **Response to the following comments of Reviewer K6MD**
> >
> > Dear Reviewer **K6MD**:
> >
> > Thank you for your constructive follow-up and for acknowledging the majority of our points in the initial rebuttal. We sincerely appreciate your continued engagement, as your further concerns are instrumental in helping us refine our manuscript.
> >
> > We note that your remaining concerns primarily focus on the hyperparameter sensitivity within Eqs. 4 and 5 of our MWAM. First and foremost, we wish to emphasize that our intention is not to obstinately defend our method, but rather to engage in a substantive scientific discussion with you to strengthen our work.
> >
> > To address your specific points:
> >
> > 1. **On Sensitivity:** We respectfully argue that MWAM does not exhibit excessive sensitivity to these hyperparameters across different datasets and tasks. As shown in **Table 9**, the **variance in performance is well within a reasonable range**. More importantly, we successfully applied the **exact same set of hyperparameters for both the classification and segmentation tasks (as noted in Lines 370 and 418)**, which demonstrates a considerable degree of robustness. The reason we define them as hyperparameters, rather than fixed constants, is to **ensure future adaptability**—providing flexibility for potential unforeseen datasets, model architectures, or tasks that might benefit from tuning the shape of the Sigmoid function.
> >
> > 2. **On the Choice of the Sigmoid Function:** We concur that numerous excellent non-linear functions are available. Any function satisfying the three conditions outlined in our previous response (Point A3 in "Response to Reviewer K6MD's Comments (4/5)") would be a candidate. Our choice of Sigmoid was motivated by its status as one of the most standard and well-understood functions that meets our requirements. Admittedly, we did not initially anticipate the need for four hyperparameters to finely control its shape. We have already acknowledged this as a limitation in the manuscript (Lines 1348-1349) and plan to explore making these parameters learnable in our future work. We are grateful for your suggestion, which has illuminated a valuable alternative direction: reducing the number of hyperparameters. Regarding the softmax function with a temperature coefficient you mentioned, we did consider it. However, it does not satisfy our second criterion (from the previous response), which requires that the weights are not all strictly confined to be less than one. Therefore, based on our current understanding, refining the Sigmoid-based approach remains the most promising path forward for now. We look forward to exploring this topic further in subsequent research to reduce MWAM's reliance on these hyperparameters.
> >
> > 3. **On the Feasibility of Tuning and Its Benefits:** These hyperparameters are more straightforward to determine than they might appear. As per our tuning strategy (detailed in A3), they do not require an exhaustive search over a vast parameter space. Instead, they can be determined with just a few adjustments based on the desired shape of the Sigmoid function. (Perhaps this can be seen as a convenient ancillary benefit of using the well-understood Sigmoid function). Furthermore, these parameters offer a tangible advantage: they equip MWAM to handle challenging scenarios. For instance, as demonstrated in Table 10, they enhance robustness against the high noise levels encountered in online learning contexts.
> >
> > In summary, we do not deny that reducing hyperparameter dependence is a necessary stage in MWAM's evolution. However, it is a double-edged sword; the introduction of these hyperparameters currently yields tangible performance benefits, particularly in terms of robustness. We will incorporate a detailed version of this discussion into the manuscript (around Line 1175 of the supplementary material) to formally document our rationale for selecting the Sigmoid function.
> >
> > Regarding your other suggestions on methodological comparisons and the manuscript's narrative, we will revise the paper accordingly.
> >
> > Thank you once again for your time and insightful feedback. We look forward to your further reply, and wish you a pleasant day.

---

> > ### Author Response · Authors · 2025-11-17
> > **Further Response Regarding R3 and R4**
> >
> > Dear Reviewer **K6MD**:
> >
> > We sincerely thank you for your insightful and constructive feedback. Following your valuable suggestions, we have undertaken a significant restructuring of the manuscript's main body and experimental tables to improve its clarity and logical flow.
> >
> > First, to better integrate our theoretical and experimental evidence into the main narrative, we have introduced a new Chapter 3 in the manuscript. This new section now houses key proofs and supporting experiments. To make this possible, we utilized the additional pages permitted during this rebuttal stage. Furthermore, we have carefully revised the text within this chapter to strengthen the cross-references to related, more detailed derivations that remain in the appendix.
> >
> > Second, we have thoroughly reorganized the presentation of Table 2 and Table 16. The specific changes include:
> >
> > 1. The results for the vanilla GSS have been moved from Table 16 to Table 2, while the LS3M results have been moved from Table 2 to Table 16. This groups the results more logically.
> >
> > 2. We have also enhanced the narrative linkage between these two tables to make their relationship clearer to the reader.
> >
> > 3. Crucially, for improved readability and direct comparison, we have now arranged the vanilla methods and their corresponding +MWAM-enhanced versions side-by-side.
> >
> > The revised version of the manuscript reflecting all these changes has been uploaded to the system. We believe these revisions have substantially improved the paper and are eager to receive your feedback.
> >
> > Sincerely,
> >
> > authors of #2549.

---

> ### Comment · Reviewer_K6MD · 2025-11-19
>
> Thank you for the detailed clarifications and the additional explanations provided in the rebuttal. I appreciate the authors’ efforts to support the empirical robustness of the hyperparameters and the novelty of adopting a frequency-domain perspective for handling missing modalities and modality imbalance. Nevertheless, after carefully revisiting the paper and the rebuttal (as well as the other reviewers’ comments), several fundamental concerns still remain unaddressed, and further concerns have emerged, as outlined below:
>
> &nbsp;
>
> ### **1.Insufficient Justification for Low-Frequency Dominance**
> The central premise, that modalities with higher low-frequency energy inherently dominate optimization, lacks adequate theoretical and empirical support to justify its generality. While spectral bias is well documented in prior works, it still requires that this assumption reliably reflects semantic importance across different modalities or tasks. In applications where high-frequency information is critical (e.g., edge- or detail-sensitive recognition), the formulation may fail, yet such limitations are not examined. This means that relying solely on the Frequency Ratio Metric (FRM) to judge modality importance appears to be a dangerous generalization and excessive. Consequently, FRM as a proxy metric appears to overgeneralize spectral bias without rigorous validation.
>
>
> ### **2. Weak Alignment Between Theory and Method**
> The connection between the presented theorems and the proposed approach remains unclear. The theorems describe broad optimization tendencies (e.g., spectral bias) rather than establishing a specific theoretical hypothesis that necessitates the architecture. As a result, the theoretical section reads more like a post-hoc characterization of observed behavior than a principled foundation driving the method.
>
>
> ### **3.Heavy Reliance on Heuristic Hyperparameter Tuning**
> This part still remains. The need for a reflected sigmoid with four hyperparameters ($\alpha, \beta, \gamma, \lambda$) suggests that FRM alone does not provide a sufficiently stable or discriminative signal for modality weighting. The authors’ own remark, that “did not initially anticipate the need for four hyperparameters”, further indicates that this component emerged from empirical tuning rather than theoretical grounding, highlighting the gap between the stated motivation and the actual implementation.
>
>
> ### **4. Lack of Clarity and Coherence in Writing**
> The overall structure and narrative flow of the paper are not smooth. The transition from theoretical discussions to empirical analysis and then to the proposed method lacks coherence, making it difficult to follow the logical progression of the research. In particular, the exposition lacks standard mathematical structure (e.g., clearly stated theorems, proofs, and formal derivations), which further weakens the clarity of the presentation.
>
> &nbsp;
>
> In summary, while the frequency-domain perspective is novel and the empirical results are encouraging, the main claims, methodological justification, and overall presentation are not sufficiently established. From my perspective, the level of rigor is still inadequate, and I therefore believe the paper is not yet ready for acceptance at this stage.

---

> > ### Author Response · Authors · 2025-11-19
> > **Response to Reviewer K6MD**
> >
> > We must express our profound confusion regarding the latest set of comments. We engaged in this discussion with the sincere intention of respecting your feedback and improving our manuscript. However, we are taken aback by the rush to a conclusive judgment, especially when it appears to be based on significant misunderstandings and an oversight of critical information already present in our manuscript and rebuttal.
> >
> > **1)Regarding the claim that our work “lacks adequate theoretical and empirical support to justify its generality” and that its limitations in high-frequency applications “are not examined.”**
> >
> > We are perplexed by this assertion, particularly in light of your statement about “carefully revisiting the paper and the rebuttal.” Had this been the case, our extensive discussion in Appendix A.11 on fine-grained classification could not have been overlooked. As is well-established, fine-grained classification is a quintessential high-frequency dependent task. In this section, we not only discuss this exact scenario but also provide empirical evidence demonstrating our method's effectiveness. This directly refutes the claim that “the formulation may fail, yet such limitations are not examined.” The limitation was indeed examined, and the formulation was shown to succeed. Therefore, the summary judgment that “FRM as a proxy metric appears to overgeneralize spectral bias without rigorous validation” is demonstrably false, as the rigorous validation exists within the appendix you claim to have carefully reviewed. Was this section intentionally ignored?
> >
> > **2)Regarding the assertion that “the theoretical section reads more like a post-hoc characterization of observed behavior than a principled foundation driving the method.”**
> >
> > This comment suggests a fundamental misinterpretation of the scientific methodology. Principled research often follows the exact path we took: (i) We first identified an empirical phenomenon through experiments—that modal preference can be characterized in the frequency domain. (ii) To formalize this observation, we then developed the theoretical proofs in Appendix A.1, which in turn provided the principled foundation for designing the FRM.
> >
> > This is not a “post-hoc characterization”; it is the standard process of developing theory from empirical evidence. The alternative you seem to suggest—finding an abstract theory first and then searching for a computer vision problem it might solve—is contrary to how impactful, problem-driven research is conducted.
> >
> > **3)Regarding the critique of “Heavy Reliance on Heuristic Hyperparameter Tuning.”**
> >
> > This criticism is based on two apparent misunderstandings. First, our statement that we “did not initially anticipate the need for four hyperparameters” reflects a normal, progressive research process. To ask if any model is designed perfectly in a single iteration is to ignore the reality of scientific discovery. Second, and more critically, your re-review seems to have missed the core architecture: the FRM and the weight allocation function are two distinct, cooperative components. The FRM evaluates modal preference, and the function translates that evaluation into weights. We have never claimed they provide independent gains. The hyperparameters exist within the allocation function to ensure its adaptability to diverse tasks and harsh training environments. For all standard tasks in the main paper, a single, fixed set of hyperparameters was used, and their insensitivity was demonstrated in Appendix A.9. This was explicitly mentioned in our rebuttal to reviewer bfT5. We are concerned this was also missed during your “careful” re-evaluation.
> >
> > **4)Regarding the comment that “The overall structure and narrative flow of the paper are not smooth.”**
> >
> > This comment is deeply troubling because it criticizes the very change you previously advocated for. Our initial submission followed a clear “problem-method-validation” structure, which received positive feedback from reviewer gN31.In the previous round, you recommended moving theoretical content from the appendix to the main text. We explicitly expressed concern that this would “disrupt the article's structure” but implemented a version of this change in the rebuttal pages out of respect for your suggestion. Now, you are criticizing the very result of that change. We cannot comprehend the reason for this direct contradiction. Fortunately, we have preserved the original manuscript structure, which allows us to easily revert the changes made solely to accommodate your previous feedback.
> >
> > We remain deeply concerned by the contradictory nature of this review and the apparent disregard for key evidence provided in our manuscript and rebuttal. We welcome a genuine, evidence-based discussion to resolve these profound misunderstandings. We apologize if our tone in this response appears overly direct, but we believe such clarity is the most appropriate way to address the serious inaccuracies in your latest comments.

---

> ### Comment · Reviewer_K6MD · 2025-11-19
>
> Thank you for your detailed response. I appreciate the effort you put into addressing the concerns. I would like to clarify that my intention was not to undermine the work, but to discuss proposed method from my perspectives.
>
> 1. Regarding the first justification for using FRM, you mentioned that it is related to “low frequency.” My concern is that if low frequency is not dominant in a particular modality, FRM may not be the most appropriate metric. Even in Appendix A.11, the authors assume that fine-grained datasets are highly related to high-frequency components. In my view, this assumption should be either empirically validated or supported by citations from previous work, rather than stated without evidence. Specifically, it would be valuable to quantify the proportion of low- and high-frequency components in the datasets and examine how these proportions affect the proposed methods. Including such comparisons could help resolve this concern.
>
> 2. I apologize for incorrectly using the term “post-hoc.” My main point is that theoretical arguments should be formulated starting from the foundational problem and then developed into the corresponding solution. In the current manuscript, phrases such as “tends to dominate” or “tend to prioritize” appear in the Theorems section without accompanying mathematical statements, which makes them insufficiently rigorous for formal theorem presentation. This is why I had the impression that the theorems may have been constructed after the solution was proposed. I also feel that expressions like “misunderstanding scientific methodology” are unnecessarily strong for response, so I hope this can be toned down in the future.
>
> 3. I understand that reviewers may hold different perspectives. While the authors mentioned to reviewer bfT5, I still have concerns. Having one or two additional loss coefficients (e.g., $\lambda_1 \mathcal{L}_1 + \lambda_2 \mathcal{L}_2$) is generally acceptable, as these mainly involve simple scaling. However, introducing 4 or 5 method-specific hyperparameters makes it difficult to determine whether the performance gains originate from the core idea or from extensive tuning. Table 9 already shows variations of about $\pm 1.3$\% and a more thorough ablation, such as fixing each hyperparameter individually, would help clarify sensitivity. Without such analysis, the robustness and originality of the approach become harder to assess.
>
> 4. If my previous comments caused any confusion, I sincerely apologize. However, when I suggested “moving theoretical content from the appendix to the main text,” my intention was not simply to relocate the material. My recommendation was to include at least a concise summary or the key theoretical results in the main text, while leaving the detailed derivations in the appendix.
> Furthermore, as I mentioned during the rebuttal, I suggested that important equations and conceptual statements be clearly structured as Definitions, Lemmas, or Theorems (as exemplified in Appendix A.1), in order to enhance clarity and ensure that the logical flow of the theoretical arguments is easy to follow.
> My concern is that the current “theorems” in the main text read more like hypotheses or mathematically self-evident / trivial statements rather than formal theorems. Although the authors include some proofs, there remains a substantial gap between a standard, rigorous theorem–proof structure and the form in which the results are currently presented. This makes it difficult to interpret the claims as formal theoretical guarantees in the conventional mathematical sense.
>
> &nbsp;
>
> Extending point 3, I believe the number of additional hyperparameters is closely tied to assessing “novelty.” Standard hyperparameters (learning rate, batch size, weight decay, etc.) are expected and do not affect conceptual clarity, but method-specific hyperparameters should remain minimal unless essential and theoretically justified. In this regard, Occam’s Razor applies: a simpler method with fewer assumptions and tunable components is generally preferable. Excessive hyperparameters risk obscuring the core idea and weakening the conceptual contribution. I also clarify that this reflects my general philosophy on method design, independent of my evaluation of this paper.
>
> I acknowledge that some of the responses were strong, but I appreciate the authors’ efforts to engage in the discussion. This is fully acceptable to me, as my intention has been to provide constructive suggestions and raise substantive concerns. I see considerable potential in this work and do not intend to reject it; however, I believe further revisions are necessary to strengthen the contribution. I believe that the AC and PC will carefully consider both the authors’ clarifications and my review in their final decision.
>
> I am welcome any additional challenges or critical feedback on my points.

---

> ### Author Response · Authors · 2025-11-19
> **Response to Reviewer K6MD**
>
> Thank you for your thoughtful engagement and for clarifying your perspective. We sincerely appreciate the constructive dialogue we have established and are keen to continue our discussion.
>
> **I. Regarding the first point in your comments, we would like to offer two clarifications:**
>
> 1. **On the motivation for FRM's design:** The general conclusion that single-modality vision-based neural networks exhibit a preference for low-frequency information is not an unsubstantiated claim but is well-grounded in the literature (e.g., [1, 2]). Furthermore, it is important to note that while learning, these networks do not rely solely on low-frequency components; high-frequency information is also crucial for their performance, as evidenced by studies such as [3, 4].
>
> 2. **On Fine-Grained Visual Categorization (FGVC):** Before proceeding, we believe it is essential to establish two points of consensus with the reviewer. First, the premise that FGVC tasks are highly correlated with high-frequency components is not a baseless assumption. High-frequency energy typically corresponds to textures and fine details, which are particularly vital for distinguishing between image classes with subtle variations. In other words, fine-grained differences often manifest in local textures and edge details—regions rich in high-frequency information. This principle is supported by a large body of existing work, such as [5]. Second, it is a well-established principle that in any natural, unaltered image, the energy of low-frequency components vastly exceeds that of high-frequency components, as the energy distribution across the frequency domain follows a Power Law. In the context of FGVC, this means that the low-frequency information of images (e.g., shapes, outlines) is often highly similar across different sub-categories, making them less discriminative [6].
>
> To summarize these two points of consensus: **First**, the vast majority of energy in any natural image is concentrated in the low-frequency bands, making them dominant in terms of energy. **Second**, despite their lower energy, high-frequency components carry highly discriminative information for FGVC, whereas the dominant low-frequency information offers little distinguishing power due to its similarity across classes.
>
> Building upon this established consensus, we extend our reasoning from FGVC to a broader class of visual understanding tasks where high-frequency information is a dominant factor. Imaging modalities with a very high FRM lack critical high-frequency details, which are precisely the discriminative information (e.g., textures) required for FGVC. However, deep neural networks exhibit a "spectral bias" [1], tending to prioritize learning simpler, lower-frequency patterns first. Therefore, a high FRM suggests that a modality is more likely to cause the model to overfit to global shapes (low-frequency cues) while failing to capture the fine details essential for the task. Our method addresses this by assigning lower weights to high-FRM modalities, thereby mitigating this bias and encouraging the model to focus on low-FRM modalities. As we state in our manuscript (Appendix A.8), a direct or weighted summation of low- and high-frequency components would risk drowning out the crucial contributions of the high-frequency information.
>
> In addition, we have undertaken substantial revisions to the manuscript. These changes go beyond textual improvements and include new experimental results to further strengthen our work. Specifically, we have made the following key modifications:
>
> **First**, to improve the organizational flow and readability of the appendix, we have swapped the positions of Section A.11 and Section A.16.
>
> **Second**, we have significantly enhanced the section on fine-grained classification experiments (now located in the revised Section A.16). This enhancement serves two main purposes: (1) We experimentally demonstrate that fine-grained classification is a task highly dependent on high-frequency components. This is achieved by processing the dataset with both high-pass and low-pass filters of varying kernel sizes. (2) We now provide a comparative analysis of our method against the baseline under these conditions. These new results further validate the feasibility and superiority of our proposed methodology.
>
> [1] Qin et al. Overview frequency principle/spectral bias in deep learning, Communications on Applied Mathematics and Computation, 2025.
>
> [2] Xu et al. Learning in the Frequency Domain, CVPR,2020.
>
> [3] Geirhos et al. ImageNet-trained CNNs are biased towards texture; increasing shape bias improves accuracy and robustness, ICLR, 2019.
>
> [4] Wang et al. High Frequency Component Helps Explain the Generalization of Convolutional Neural Networks, CVPR, 2020.
>
> [5] Wang et al. High-frequency Component Helps Explain the Generalization of Convolutional Neural Networks, CVPR,2020.
>
> [6] Wei et al. Fine-Grained Image Analysis with Deep Learning: A Survey, Tpami,2021

---

> ### Author Response · Authors · 2025-11-19
> **Response to Reviewer K6MD**
>
> **III. Regarding Comments 2 and 4:**
>
> We would like to sincerely thank you for your patient and detailed explanation of your perspective, especially given the strong wording of our previous response. We begin by offering our sincere apologies for any language that may have come across as overly strong. We are confident that we share a common goal: to maximize the manuscript's contribution to the community.
>
> After carefully considering your latest comments, we have undertaken a significant revision of the manuscript. Key improvements include:
>
> Structural Refinement: The main theorems are now presented directly in the main body for clarity, with their complete proofs provided in the appendix.
>
> Rigor and Formality: We have paid special attention to the rigor of our wording throughout the paper to ensure the argumentative structure is more formal and precise.
>
> Coherence: The overall narrative and logical flow of the paper have been substantially improved.
>
> We believe the revised version is a marked improvement over the previous one, and this is in large part due to your constructive efforts. The updated manuscript has been uploaded to the system.
>
> We look forward to your further feedback.

---

> ### Author Response · Authors · 2025-11-19
> **Response to Reviewer K6MD**
>
> **II. Regarding the third point in your comments:** We believe it may not be necessary to repeatedly clarify the motivation and function of the weight allocation function with four hyperparameters. In fact, the table in Appendix A.9 already contains the experimental results you suggested. We acknowledge, however, that the original presentation of the table may not have been sufficiently clear.
>
> To address this, we have reorganized the table to present the results more transparently. The methodology for our sensitivity analysis was to vary one hyperparameter within a controlled range while keeping the other three fixed. In this round of discussion, we are providing the reorganized table first and are prepared to conduct further experiments if they are still deemed necessary after your review.
>
> |Params.|Acc.|PCR|
> |-|:-:|:-:|
> |1.2/1.0/0.7/6.0|96.48|3.34|
> |**1.5/1/0.7/6.0**|97.03|2.02|
> |1.8/1.0/0.7/6.0|96.42|3.09|
> |----------------|-----|-----|
> |1.5/1.3/0.7/6.0|96.04|3.45|
> |**1.5/1.0/0.7/6.0**|97.03|2.02|
> |1.5/0.7/0.7/6.0|96.53|2.32|
> |----------------|-----|-----|
> |1.5/1.0/1.0/6.0|96.73|2.57|
> |**1.5/1.0/0.7/6.0**|97.03|2.02|
> |1.5/1.0/0.4/6.0|96.37|3.30|
> |----------------|-----|-----|
> |1.5/1.0/0.7/6.3|96.07|3.51|
> |**1.5/1.0/0.7/6.0**|97.03|2.02|
> |1.5/1.0/0.7/5.7|95.87|3.26|
> |----------------|-----|-----|
> |2.0/1.5/1.3/6.5|97.08|2.57|
> |**1.5/1.0/0.7/6.0**|97.03|2.02|
> ---
>
> Given the importance of the technical points discussed above, we have prioritized this response to address them first. We would be happy to discuss the specific textual revisions in a subsequent round of discussion.
>
> In fact, we argue that Batch Normalization (BN)[1] is driven by a similar design rationale. BN introduces two learnable affine parameters (scale and shift) primarily to mitigate the issue where inputs could drive activation functions, such as Sigmoid, into their saturation regions.
>
> Our approach is analogous in principle, but with two key distinctions. First, our transformation involves both shifting and scaling along two dimensions (the x and y axes). Second, and more critically, we deliberately fix these four parameters based on statistical principles, rather than making them learnable during training.
>
> We have explicitly acknowledged this design choice in the Limitations section of our manuscript. Furthermore, we have identified the exploration of learnable parameters as a significant and promising direction for our future work.
>
> [1] Sergey Ioffe et al. Batch Normalization: Accelerating Deep Network Training by Reducing Internal Covariate Shift, ICML, 2015.

---

### Official Review · Reviewer_bfT5 · 2025-11-01

**Soundness:** 2
**Presentation:** 3
**Contribution:** 2
**Rating:** 6
**Confidence:** 3

**Summary:**

This paper tackles robustness in multimodal image understanding, focusing on cases where some modalities are missing at inference. The authors show that models often become biased toward dominant modalities, causing severe performance drops when those are absent. To address this, they propose Frequency Ratio Metric (FRM) and Multimodal Weight Allocation Module (MWAM). Experiments across multiple tasks and architectures demonstrate that MWAM improves robustness and outperforms several state-of-the-art methods.

**Strengths:**

1) Using frequency domain analysis (FRM) to diagnose and mitigate modality bias sounds interesting, which goes beyond spatial domain balancing.
2) The proposed MWAM is a lightweight and plug-and-play module with negligible computational overhead and no additional parameters during inference. This makes it attractive for real-world deployment.
3) The method is validated on multiple tasks (classification, segmentation, detection) and datasets, showing improvements over baseline  methods.  Extensive ablation studies on FRM design, window size, parameter sensitivity, and training setting are ablated.

**Weaknesses:**

1. The approach is tailored to image-based modalities and frequency domain analysis. It is unclear how well MWAM would generalize to other multimodal settings (e.g., audio-text, video-language) where frequency analysis may not be directly applicable
2. MWAM introduces several hyperparameters that require careful tuning for different tasks and batch sizes. While sensitivity analysis is provided and the authors claim the performance is not that sensistive to such hyper-parameters, considering the performance gain over baseline method is not that significant, such performance perturbations is not neglibile. However, there is no clear guidance about how to set such hyper-parameters.

**Questions:**

1) Can MWAM and FRM be adapted for multimodal tasks involving non-image data (e.g., audio, text, tabular)? What are the challenges and potential solutions?
2) How does MWAM handle cases where modalities are highly correlated or redundant? Is there a risk of over-balancing and degrading performance?
3) Are there specific scenarios or datasets where MWAM fails to improve robustness or even degrades performance? If so, what are the underlying causes?

---

> ### Author Response · Authors · 2025-11-13
> **Response to Reviewer bfT5's Comments (1/4)**
>
> Dear Reviewer **bfT5**:
>
> Thank you for your thorough review and insightful feedback. We are pleased that you found our methodology and experimental thoroughness to be strengths of the work. We acknowledge your questions regarding specific details and are happy to provide the necessary clarifications.
>
> Next, I will help clarify the misunderstanding.
>
> **I. Summary**
>
> We focus on improving the robustness of incomplete **multi-modal visual understanding models**. We attribute the performance collapse caused by modal absence to the incomplete optimization of the internal branches of the model due to modal competition, and attempt to alleviate this collapse phenomenon by balancing the model's attention to different modalities. First, through experiments and theoretical derivations, we discover and prove that in multi-modal visual classification tasks, the low-frequency components in images will, to a certain extent, dominate the model training. Subsequently, we design FRM and MWAM to evaluate the model's preference for different modalities and adopt positive treatment methods.
>
> **II. Comments and Responses**
>
> **W1:** The approach is tailored to image-based modalities and frequency domain analysis. It is unclear how well MWAM would generalize to other multimodal settings (e.g., audio-text, video-language) where frequency analysis may not be directly applicable
>
> **R1:** We did conduct preliminary experiments on audio-visual datasets, but the performance was underwhelming. We attribute this to the core design of our method, which we refer to as MWAM. MWAM is designed as a plug-and-play module that evaluates modality dominance within the input data space. This approach is inherently sensitive to the data encoding scheme.
>
> The encoding methods for non-visual modalities, such as text and audio, are fundamentally different from those for vision. This discrepancy makes it infeasible to directly apply our FRM to assess the model's preferences for these modalities in the same manner. Acknowledging this inherent constraint, we intentionally **focused our work** and **titled** our manuscript to reflect its specific **application to the robustness of "visual understanding models."**
>
> To make this scope clearer, we will add a dedicated discussion in the "Limitations" section of our revised manuscript. We will explicitly state that the current framework is tailored for visual modalities and frame its extension to other modalities as a key direction for our future work.
>
> In fact, **we have already developed a solution to this challenge**. In a follow-up work, we explore how to balance a more diverse range of modality combinations. To summarize briefly, we recognized that the limitation of MWAM stems from its operating domain—the input data space. Therefore, in our subsequent research, we shift our perspective from the input level to the latent feature space. By evaluating changes in the latent representations of different modalities in frequency domain, we can define and address modality imbalance in a more generalizable way.

---

> ### Author Response · Authors · 2025-11-13
> **Response to Reviewer bfT5's Comments (2/4)**
>
> **W2:** MWAM introduces several hyperparameters that require careful tuning for different tasks and batch sizes. While sensitivity analysis is provided and the authors claim the performance is not that sensistive to such hyper-parameters, considering the performance gain over baseline method is not that significant, such performance perturbations is not neglibile. However, there is no clear guidance about how to set such hyper-parameters.
>
> **R2:** Thank you for your insightful comment regarding the hyperparameters. We acknowledge that we have introduced additional hyperparameters in our MWAM, specifically the four scaling factors in Equations (4) and (5).
>
> **We have addressed this point in the Appendix,** where we discuss the motivation for introducing these factors **(Lines 1151-1168)** and present a parameter sensitivity analysis **(Lines 1169-1174)**. Based on the experimental results shown in **Table 9**, we concluded that the model is not highly sensitive to these parameters, provided they are chosen within a reasonable range.
>
> As noted in the manuscript, the role of these scaling factors is **to adjust the shape of the standard Sigmoid function**. This modification enables the resulting non-linear function to allocate appropriate weights to different modalities. Here, we would like to elaborate on our empirical tuning procedure for these factors.
>
> First, to provide some context, **our goal is to adjust the shape of a Sigmoid function** where T (from Equation 5) is the independent variable. The adjusted function must satisfy the following **three requirements**:
>
> 1. **Monotonicity:** The function must be monotonically decreasing, meaning a larger Feature Regulation Module (FRM) score (indicating dominance) results in a smaller weight.
> 2. **Effective Weight Allocation:** The function must assign meaningful and distinct weights based on the T values of different modalities. These weights should not all be less than 1, as this would suppress the overall training process **(as detailed in Lines 1160-1163)**. Furthermore, since the T values are normalized and their relative differences can be small, the function needs to be sufficiently sensitive to these subtle differences to clearly reflect the model's preference for each modality, preventing the allocated weights from being influenced by noise.
> 3. **Non-Saturation:** The operating points (i.e., the T values) should fall within the approximately linear region of the Sigmoid curve, not its saturation zones, to ensure a responsive adjustment.
>
> With these requirements in mind, we employ the following two-step tuning methodology: We set the value '1' as the threshold separating dominant and weaker modalities, where dominant modalities receive weights less than 1 and weaker modalities receive weights greater than 1. These weights are then non-linearly scaled according to the differences in their respective T values.
>
> The tuning process is as follows:
>
> 1. **Initial Analysis:** We first statistically analyze the T values for each modality over a number of initial training iterations (the number depends on the dataset size). This provides a preliminary understanding of the typical numerical gap between the modalities' T values for the given task.
>
> 2. **Shape Adjustment:** We then use the four scaling factors to adjust the Sigmoid function's shape. We observe the resulting weights assigned to the modalities and tune the factors until the three requirements listed above are met.
>
> The necessity for these hyperparameters arises because different modality combinations yield different ranges and gaps in their calculated T values. The scaling factors provide the flexibility to adapt the function's output response to satisfy requirements 2 and 3 for any given scenario.
>
> In summary, by following this two-step empirical tuning method, we can effectively determine appropriate values for the scaling factors. Ultimately, all four factors serve a single, unified purpose: to reshape the standard Sigmoid function into a customized curve that is optimally suited for our modality weight allocation strategy.

---

> > ### Author Response · Authors · 2025-11-13
> > **Response to Reviewer bfT5's Comments (3/4)**
> >
> > **Q1:** Can MWAM and FRM be adapted for multimodal tasks involving non-image data (e.g., audio, text, tabular)? What are the challenges and potential solutions?
> >
> > **A1:** Thank you for your insightful comment regarding the generalization of our method. Unfortunately, to the best of our current knowledge, MWAM is not well-suited for tasks involving non-image modalities (e.g., audio). This is precisely why we intentionally scoped our work to **"Multi-Modal Image Understanding Models,"** as stated in our manuscript's title.
> >
> > To validate this, we did conduct preliminary experiments on an audio-visual classification task, but the results were suboptimal. We attribute this outcome to the core design of MWAM: it functions as a plug-and-play module that assesses modality dominance within the input data space. This approach is inherently and highly constrained by the data encoding scheme. The encoding methods for non-visual modalities like audio or text are fundamentally different from those for vision (e.g., visible light, infrared, depth maps). This discrepancy makes it infeasible to directly apply our FRM to evaluate the model's preferences for these disparate modalities.
> >
> > Acknowledging this, we will explicitly state this limitation in the "Limitations" section of our revised manuscript. We will clarify that the current framework is tailored for **image-based modalities** and frame its extension to a broader range of modalities as a key direction for our future work.
> >
> > In fact, **we are already making progress on this front.** In our follow-up research, we are exploring more diverse modality combinations by leveraging frequency domain analysis. To summarize briefly, we recognized that the limitation of MWAM stems from its operating domain—the input data space. Therefore, in our subsequent work, we shift our perspective from the input level to the latent feature space. By assessing changes in the latent representations of different modalities, we can define and address modality imbalance in a more generalizable and powerful way.
> >
> > **Q2:** How does MWAM handle cases where modalities are highly correlated or redundant? Is there a risk of over-balancing and degrading performance?
> >
> > **A2:** Thank you for these insightful comments. The two potential issues you have raised are indeed critical challenges that we meticulously considered during the design of our MWAM. We are confident that **MWAM is well-equipped with internal components to address both concerns effectively.**
> >
> > **First,** regarding the challenge of handling highly similar modalities: As elaborated in our response to **A2**, our parameter tuning strategy provides fine-grained control. Specifically, by evaluating the discrepancy 'T' between modalities (Equation 5), we can explicitly modulate the output weights of different modalities to a desired balance using the four hyperparameters in Equations 4 and 5. **This mechanism ensures that even for modalities with high superficial similarity, their weights can be precisely configured as long as their contributions to the task are not identical.**
> >
> > **Furthermore,** to handle specific outlier samples with unusually high similarity, our MWAM incorporates a component we call the **'FRM Bank'**. Governed by Equation 1, the FRM Bank is designed to identify and down-weight the influence of these anomalies, thereby ensuring the model maintains a stable and correct training trajectory.
> >
> > Crucially, the high degree of adjustability offered by the four hyperparameters is key to preventing over-balancing. This flexible control over weight allocation significantly mitigates the risk of performance degradation that could otherwise occur if the model were to excessively suppress a valuable modality.

---

> ### Author Response · Authors · 2025-11-13
> **Response to Reviewer bfT5's Comments (4/4)**
>
> **Q3:** Are there specific scenarios or datasets where MWAM fails to improve robustness or even degrades performance? If so, what are the underlying causes?
>
> **A3:** Thank you for your insightful comment. We agree that such a hypothetical scenario where our method's effectiveness is compromised is plausible, and we believe the root cause would likely be data-specific noise. We would like to address your concern **from two perspectives:** the dataset and the training scenario.
>
> 1.**From the Dataset Perspective:** We have conducted extensive validation experiments across a wide range of visual tasks and diverse datasets. These include:
>
> 1). **Multimodal Classification:** on the SURF dataset (RGB-Infrared-Depth).
>
> 2). **Semantic Segmentation:** on the NYU-Depth V2 dataset (RGB-Depth).
>
> 3). **Medical Image Segmentation:** on the BRATS2020 dataset (T1-T2-Flair-T1ce).
>
> 4). **Fine-Grained Classification:** on Stanford Dogs and FGVC-Aircraft datasets (RGB-Depth).
>
> 5). **Multimodal Object Detection:** on the DroneVehicle dataset (RGB-Infrared).
>
> Our experiments cover **modality combinations** ranging from two to four, utilize both CNN-based and ViT-based **backbones**, and span **tasks** from classification and segmentation to detection. The **application scenarios** ranged from improving the performance of vanilla models to breaking the performance bottlenecks of existing methods for handling missing modalities. The results consistently demonstrate the effectiveness of our MWAM across all these settings. While **we do not rule out the possibility that MWAM might be less effective on datasets**, modality combinations, or architectures beyond our current scope, our comprehensive experiments show a consistent performance improvement across all tested configurations.
>
> 2. **From the Training Scenario Perspective:** As we reported in the manuscript **(Lines 1176-1228)**, certain training environments can indeed affect the performance of MWAM, potentially degrading it or even rendering it ineffective. Specifically, we identified extreme scenarios like few-shot learning and online learning as having a significant negative impact. We analyzed and reported the reasons for this in **Lines 1185-1187 and 1225-1228**.
>
> To elaborate, two main factors are at play. **First**, as a plug-and-play module, MWAM's performance floor is dependent on the baseline model. **Second**, MWAM evaluates the FRM score within each mini-batch and relies on an FRM bank for stable representation updates. **Consequently,** a small mini-batch size inevitably introduces estimation errors, and this error is magnified as the batch size decreases. In an extreme case like online learning (where the mini-batch size is 1), the model evaluates dominance on a per-sample basis. If the original hyperparameters are used without modification, the model cannot effectively distinguish signal from training noise, causing the optimization path to drift and ultimately impairing performance. However, as stated in **Lines 1225-1228**, this performance loss can be mitigated by adjusting the hyperparameters.
>
> In summary, we acknowledge that specific datasets and training scenarios could exist where MWAM's performance is compromised or even nullified. However, we also believe that there are potential avenues for mitigating these effects. We plan to focus our research on these topics in the coming period.
>
> Once again, we thank you for your insightful comments. Should any part of our work or responses require further clarification, we would be happy to discuss it further with you during the rebuttal period.
>
> Best **regards**,
>
> authors of #2549.

---

### Official Review · Reviewer_gN31 · 2025-11-01

**Soundness:** 3
**Presentation:** 3
**Contribution:** 3
**Rating:** 6
**Confidence:** 3

**Summary:**

This paper addresses a common problem in multimodal learning: performance drops significantly when one of the data modalities is missing.
The authors argue this happens because the model learns to rely too much on one "dominant" modality during training, while neglecting the others. Their key idea is that this dominance can be measured in the frequency domain of the images.
To solve this, they propose:
A Frequency Ratio Metric (FRM): A simple metric to quantify how much the model prefers each modality by comparing its low-frequency and high-frequency content.
A Multimodal Weight Allocation Module (MWAM): A lightweight, plug-and-play module that uses the FRM to rebalance the training process.

**Strengths:**

This paper analyzes the modality imbalance problem from the frequency domain and introduces an FRM to quantify how much the model prefers each modality. This seems to be promising.

Based on FRM, this paper introduces MWAM to modulate the training process of multimodal models so that the performance can be balanced and enhanced.

The writting is easy to understand.

**Weaknesses:**

"The overall training process becomes more stable, as evidenced by the reduced variance in the total loss curve". But in Figure. 4 SF-MD (w/o Intervention)'s total loss is more stable than that of SF-MD (w / MD (w / Loss Intervention ntervention).

**Questions:**

Consider this paper does not utilize LLMs to perform understanding tasks, so it more like image perception work.

"SF-MD, we additionally introduce auxiliary heads, whereas we do not in MMANet." Why introduce extra heads for SFMD only?

---

> ### Author Response · Authors · 2025-11-13
> **Response to Reviewer gN31's Comments (1/2)**
>
> Dear Reviewer **gN31**:
>
> Thank you for your thorough review and insightful comments. We sincerely appreciate your positive remarks on our methodology and presentation. We also recognize your concerns about certain details and welcome this opportunity to clarify any potential misunderstandings and answer your questions.
>
> Next, I will help clarify the misunderstanding.
>
> **I. Summary**
>
> We focus on improving the robustness of incomplete **multi-modal visual understanding models**. We attribute the performance collapse caused by modal absence to the incomplete optimization of the internal branches of the model due to modal competition, and attempt to alleviate this collapse phenomenon by balancing the model's attention to different modalities. First, through experiments and theoretical derivations, we discover and prove that in multi-modal visual classification tasks, the low-frequency components in images will, to a certain extent, dominate the model training. Subsequently, we design FRM and MWAM to evaluate the model's preference for different modalities and adopt positive treatment methods.
>
> **II. Comments and Responses**
>
> **W1:** "The overall training process becomes more stable, as evidenced by the reduced variance in the total loss curve". But in Figure. 4 SF-MD (w/o Intervention)'s total loss is more stable than that of SF-MD (w / MD (w / Loss Intervention).
>
> **R1:** Thank you for your insightful comments. We would like to clarify that the observed **fluctuation is not a shortcoming of our model**. Instead, we argue that it serves as **evidence of the model's active regulation of contributions from different modalities**.
>
> As you correctly observed, the total loss of our model with the proposed loss-level intervention (SF-MD w/ Loss Intervention) exhibits a higher degree of fluctuation compared to the vanilla SF-MD (w/o Intervention). We believe this is an expected and normal phenomenon, which can be explained from the following **three perspectives**.
>
> **First**, the total loss in the vanilla SF-MD is merely a **sub-component of the total loss** in our proposed method, which naturally leads to a smoother curve for the former. Specifically, we design auxiliary training heads for each modality to evaluate their individual classification performance. Based on these single-modality logits, we compute a separate Cross-Entropy (CE) loss for each. These individual losses are then combined in a weighted sum with the original CE loss, guided by the results from our FRM. We posit that this loss weighting acts as a global intervention mechanism. Consequently, the introduction of these new loss sub-terms inevitably perturbs the trajectory of the total loss. Since these sub-terms are absent in the vanilla SF-MD, its total loss curve is naturally smoother. Crucially, as shown in Table 5, the results demonstrate that this strategy of intervening at the loss level is undeniably beneficial for the model's final performance.
>
> **Second**, in the vanilla SF-MD (w/o Intervention), the losses from different modalities are intertwined, preventing the model from identifying and focusing on weaker modalities that are suppressed by a dominant one. These weaker modalities are precisely the ones that require more attention. As a result, they are persistently overshadowed, failing to learn useful representations. This leads to a significant performance collapse when the dominant modality is absent. In contrast, our SF-MD (w/ Loss Intervention) can **identify the dominant modality** (e.g., the depth map) and strategically shift its focus to learning from the other modalities. The single-modality losses are no longer entangled, indicating that the model is not greedily learning from the dominant modality alone but is also ensuring the learning of the others.
>
> **Finally**, this loss fluctuation is mitigated after introducing our mixing strategy. This is because the model then considers not only the global-level regulation (via the loss function) but also a more fine-grained adjustment (via gradient modulation of individual encoders). Therefore, the fluctuation can be effectively alleviated and is well-justified within our complete framework.
>
> In summary, we respectfully argue that **the observed fluctuation is not a flaw**. Rather, it is tangible evidence that our model is **successfully regulating the contributions of different modalities**—a core objective of our proposed intervention.

---

> ### Author Response · Authors · 2025-11-13
> **Response to Reviewer gN31's Comments (2/2)**
>
> **Q2:** "SF-MD, we additionally introduce auxiliary heads, whereas we do not in MMANet." Why introduce extra heads for SFMD only?
>
> **A2:** Thank you for your valuable comment. The decision to introduce the auxiliary classification heads solely within the SF-MD framework was a deliberate choice to rigorously evaluate the effectiveness of our loss intervention strategy in isolation.
>
> Specifically, SF-MD represents the most "vanilla" architecture in our study. Applying the intervention here allowed us to observe its direct impact in a clean and controlled setting, without confounding factors. MMANet, on the other hand, is a more advanced model built upon SF-MD, incorporating its own sophisticated strategies to identify dominant modalities and refine decision boundaries.
>
> The purpose of the auxiliary heads is to enable intervention at the loss level during training. Since MMANet already employs a two-stage training process and includes additional loss sub-terms, we were initially concerned that introducing another loss-level intervention could disrupt its carefully designed optimization process for the decision boundary, potentially leading to performance degradation. Therefore, we adopted a more conservative and targeted strategy for MMANet: modulating the gradients of its internal encoders.
>
> However, inspired by your insightful suggestion, we have now conducted the corresponding experiment. **We applied the loss intervention strategy to MMANet**, effectively creating a hybrid intervention approach. The results are presented in **Table 1**.
>
>   Table 1. Performance of MMANet with Different MWAM Strategies
>
> |Modalities|||MMANet||MMANet+MWAM(w/ Gradient)|| MMANet+MWAM(w/ Hybrid Loss+Grad )||
> |:-:|:-:|:-:|:-:|:-:|:-:|:-:|:-:|:-:|
> |Rgb|Depth|Ir|Acc.|PCR|Acc.|PCR|Acc.|PCR|
> |√|×|×|91.43|7.77|93.49|5.32|91.80|7.50|
> |×|√|×|97.73|1.41|98.31|0.44|98.19|1.06|
> |×|×|√|89.96|9.25|94.18|4.62|94.10|5.18|
> |√|√|×|98.39|0.75|98.64|0.10|98.80|0.44|
> |√|×|√|96.99|2.16|96.60|2.17|97.58|1.67|
> |×|√|√|98.82|0.31|99.27|-0.54|99.14|0.10|
> |√|√|√|99.13|/|98.74|/|99.24|/|
> |Average|||96.06|3.61|**97.03**|**2.02**|96.98|2.66|
> ---
>
> The experimental results indicate that after incorporating the weighted loss intervention, MMANet exhibits slight performance fluctuations compared to its baseline counterpart without this strategy. This observation suggests that the weighted loss does indeed influence MMANet's inherent mechanism for adjusting its decision boundary. Nevertheless, the model still outperforms the vanilla MMANet, which ultimately validates the effectiveness of our MWAM-guided intervention strategy.
>
> Once again, we thank you for your insightful comments. Should any part of our work or responses require further clarification, we would be happy to discuss it further with you during the rebuttal period.
>
> Best **regards**,
>
> authors of #2549.

---

### Official Review · Reviewer_Lu1c · 2025-11-03

**Soundness:** 3
**Presentation:** 2
**Contribution:** 3
**Rating:** 6
**Confidence:** 3

**Summary:**

This paper tackles the problem of modality bias and missing modality robustness in multimodal image understanding models. The authors observe that current multimodal systems tend to overfit or over-rely on certain “dominant” modalities (e.g., depth or RGB), leading to severe performance drops when one modality is missing at inference time. To address this, the paper introduces Frequency Ratio Metric and Multimodal Weight Allocation Module. Extensive experiments on segmentation (BRATS2020, NYU-Depth V2) and classification (CASIA-SURF) benchmarks demonstrate that MWAM consistently improves robustness and performance across CNN- and ViT-based architectures with minimal computational overhead.

**Strengths:**

- he idea of diagnosing and correcting multimodal imbalance in the frequency domain is both intuitive and underexplored. The FRM formulation provides a new angle that complements existing spatial-domain balancing techniques.
- MWAM is architecture-agnostic, parameter-light, and easy to integrate into existing backbones. This makes it particularly attractive for practitioners working on robustness in multimodal models.
- The authors validate their method on diverse datasets and tasks (segmentation and classification) and show consistent improvements in both accuracy (Dice, MIoU, Acc) and robustness (PCR).
- The ablation studies (gradient vs. loss vs. hybrid intervention) clearly show why hybrid balancing works best. The training loss visualizations (Fig. 4) effectively illustrate MWAM’s stabilizing effect.

**Weaknesses:**

- While the frequency-domain motivation is intuitive, the paper lacks a rigorous theoretical connection between FRM and gradient dynamics. A more formal justification for why FRM effectively measures modality dominance would strengthen the contribution.
- All experiments are on moderate-sized datasets. It remains unclear whether FRM and MWAM scale effectively to modern multimodal foundation models.
- Some figures and equations are dense and could be better formatted for readability.

**Questions:**

- How sensitive is the FRM metric to the choice of frequency cutoff (p × p patch size and q × q block size)? Could adaptive frequency selection further improve the metric?
- How does MWAM compare to feature-level reweighting techniques like modality dropout, dynamic fusion gates, or attention-based balancing in terms of convergence stability?
- Would FRM or MWAM also help in text–image or video–audio multimodal models, or is the current formulation specific to visual modalities only?

---

> ### Author Response · Authors · 2025-11-13
> **Response to Reviewer Lu1c's Comments (1/2)**
>
> Dear Reviewer **Lu1c**:
>
> Thank you for your thorough review and constructive feedback on our manuscript. We appreciate your recognition of the strengths in our research perspective, methodology, and experiments. We acknowledge your questions regarding specific details and are happy to provide the necessary clarifications.
>
> Next, I will help clarify the misunderstanding.
>
> **I. Summary**
>
> We focus on improving the robustness of incomplete **multi-modal visual understanding models**. We attribute the performance collapse caused by modal absence to the incomplete optimization of the internal branches of the model due to modal competition, and attempt to alleviate this collapse phenomenon by balancing the model's attention to different modalities. First, through experiments and theoretical derivations, we discover and prove that in multi-modal visual classification tasks, the low-frequency components in images will, to a certain extent, dominate the model training. Subsequently, we design FRM and MWAM to evaluate the model's preference for different modalities and adopt positive treatment methods.
>
> **II. Comments and Responses**
>
> **W1**: While the frequency-domain motivation is intuitive, the paper lacks a rigorous theoretical connection between FRM and gradient dynamics. A more formal justification for why FRM effectively measures modality dominance would strengthen the contribution.
>
> **R1**: Thank you for your insightful comments. We particularly appreciate your suggestion regarding the need for theoretical support.
>
> Regarding this point, we would like to gently draw your attention to **Appendix A.1 (Lines 739-847)**, where we have provided a detailed theoretical proof. We believe this section may contain the justification you are looking for, and we would be grateful for your confirmation.
>
> To briefly summarize our core idea and its theoretical foundation: Our research is built upon a key observation: the performance collapse of multi-modal models when a modality is missing stems from the model's inherent preference for certain modalities. We posit that this preference can be quantified in the frequency domain. Our proposed FRM is a direct method to measure this preference, defined as the ratio of **low-frequency components to their high-frequency counterparts within each modality**.
>
> The theoretical derivations in **Appendix A.1 provide the formal support for this concept**. From the perspective of optimization dynamics, we deduce how intra-modal low-frequency components influence the optimization of this modal preference. This analysis serves as the necessary theoretical underpinning for our FRM. Due to page limitations in the main paper, we placed this detailed derivation in the appendix, where it serves as supplementary evidence for the design motivation of FRM.
>
> If this is not the theoretical justification you were anticipating, or if you have further questions, we would be more than happy to engage in a more in-depth discussion to address your concerns.
>
> **W2:** All experiments are on moderate-sized datasets. It remains unclear whether FRM and MWAM scale effectively to modern multimodal foundation models.
>
> **R2:** Thank you for this excellent suggestion. This represents a substantial undertaking, as it requires a thorough search for suitable datasets followed by a comprehensive set of new experiments. We agree that evaluating our model on even larger-scale datasets would further strengthen our findings. To this end, we are actively searching for suitable, larger-scale, image-based multi-modal datasets for additional training and evaluation. We plan to report any potential results from these experiments in a subsequent update. We have already initiated this process. However, given the time-intensive nature of these experiments, we would like to submit our responses to your other comments first to ensure a timely progression of the review process.
>
> We would be extremely grateful for any recommendations on datasets that you believe would be particularly suitable for this evaluation. Any suggestions would be highly valuable to us.
>
> **W3:** Some figures and equations are dense and could be better formatted for readability.
>
> **R3:** Thank you for pointing this out. You have accurately identified one of the most difficult dilemmas we encountered while writing this paper: the trade-off between thoroughness and brevity. We endeavored to provide a comprehensive account, but struggled to do so within the given space constraints.

---

> ### Author Response · Authors · 2025-11-13
> **Response to Reviewer Lu1c's Comments (2/2)**
>
> **Q1:** How sensitive is the FRM metric to the choice of frequency cutoff ($p \times p$ patch size and $q \times q$ block size)? Could adaptive frequency selection further improve the metric?
>
> **A1:** Thank you for this very insightful comment. Regarding the selection of $p$ and $q$, we have conducted a parameter sensitivity analysis, which is detailed in **lines 1026 to 1063 of our manuscript**. The results of this analysis concluded that the **model exhibits low sensitivity to these two parameters**, as the performance variations were consistently within a manageable range of variance. Based on a comprehensive evaluation of these experiments, we determined the optimal settings to be $p=8$ and $q=2$.
>
> Furthermore, we concur with your perspective that making $p$ and $q$ adaptive could, in theory, further enhance the model's performance. Such a mechanism would allow the model to dynamically adjust its scaling in response to the varying scales of targets in the input. However, we believe that implementing this concept effectively would require a rigorous architectural redesign and substantial experimental validation. It would be crucial to unequivocally demonstrate the effectiveness and superiority of such a novel structure for this class of tasks.
>
> Therefore, given its significant potential and the careful investigation it warrants, we have prioritized this direction as a key focus for our future work.
>
> **Q2:** How does MWAM compare to feature-level reweighting techniques like modality dropout, dynamic fusion gates, or attention-based balancing in terms of convergence stability?
>
> **A2:** Thank you for this insightful comment. We appreciate you raising this point and would like to clarify that a comprehensive comparison with the types of methods you mentioned is indeed presented in our manuscript.
>
> Specifically, **Table 18 (lines 1448-1457)** benchmarks the performance of our MWAM against several representative SOTA strategies. To be more explicit, these methods cover the exact categories of interest:
>
> 1. **CMRT-JT**, which is a modality preference-based method.
> 2. **DCP**, which is a prompt-based method.
> 3. **COM**, which is an attention mask-based method.
> 4. **A2FSeg** (in Table 16), which is an adaptive modal discarding method
>
> Furthermore, the comparison with modality dropout-based approaches is also included. The **SF-MD** method, presented in **Table 3 and Table 4** of the main text, represents this particular strategy.
>
> As the experimental results in these tables demonstrate, our proposed MWAM consistently outperforms all these competing SOTA strategies, achieving the best performance.
>
> **Q3:** Would FRM or MWAM also help in text–image or video–audio multimodal models, or is the current formulation specific to visual modalities only?
>
> **A3:** Thank you for your comment. We agree that our method is tailored for **vision-based modalities**. This was a deliberate design choice, and it forms the core of our contribution.
>
> Our paper's scope is explicitly defined as **multi-modal image understanding**, a point we stress in the **title** and throughout the manuscript. Therefore, the applicability to **vision-based tasks** is not a secondary outcome but the central focus of our investigation.
>
> While we clearly define this scope, we also recognize its boundaries. As mentioned in our "Limitations" section, extending this work to more diverse modalities is a primary goal for **our future research**. We are pleased to report that we are already well underway with a more generalizable solution. Our next-generation framework moves beyond the input data space—the main constraint of the current method—and operates in the latent feature space. By analyzing the dynamics of latent representations, we can address modality imbalance in a manner that is fundamentally agnostic to the initial data encoding, making it suitable for a much broader set of modalities.
>
> Once again, we thank you for your insightful comments. Should any part of our work or responses require further clarification, we would be happy to discuss it further with you during the rebuttal period.
>
> Best **regards**,
>
> authors of #2549.

---

> ### Author Response · Authors · 2025-11-19
> **Performance results on larger datasets**
>
> Following the reviewer's valuable suggestion, we have conducted extensive experiments on a larger-scale dataset to further validate the scalability and robustness of our proposed method.
>
> For this purpose, we selected the CASIA-CeFA dataset, a large-scale, multi-modal, and cross-ethnicity face anti-spoofing dataset. It comprises 766k image triplets, each containing an RGB, a Depth, and an Infrared (IR) image. We strictly adhered to the official data partitioning Protocol 3, utilizing 216k samples for training, 110k for validation, and 438k for testing. The experimental results are presented in the table below.
>
> ---
> |Modalities|||SF-MD||MMANet||SF-MD+MWAM||
> |:-:|:-:|:-:|:-:|:-:|:-:|:-:|:-:|:-:|
> |Rgb|Depth|Ir|Acc.|PCR|Acc.|PCR|Acc.|PCR|
> |√|×|×|97.88|2.12|98.00|2.00|98.55|1.45|
> |×|√|×|99.96|0.04|99.98|0.02|99.99|0.01|
> |×|×|√|99.71|0.29|99.80|0.20|99.80|0.2|
> |√|√|×|99.97|0.03|99.98|0.02|99.97|0.03|
> |√|×|√|99.95|0.05|99.97|0.03|99.99|0.01|
> |×|√|√|100|0|99.99|0.01|100|0|
> |√|√|√|100|/|98.74|/|100|100|/|
> |Average|||99.63|0.42|99.67|0.38|99.76|0.28|
> ---
>
>
> Our analysis of these new results reveals several key insights. The substantial increase in dataset size enabled the models to learn more comprehensive and robust representations. Specifically:
>
> In the large dataset, SF-MD demonstrates a significant improvement in prediction accuracy by effectively learning to represent and compensate for missing modalities.
>
> Building upon SF-MD, MMANet achieves a further performance gain by refining the decision boundaries.
>
> Crucially, when equipped with our proposed MWAM module, SF-MD further boosts its discriminative power by adaptively adjusting the model's modality preferences.
>
> In summary, these experiments compellingly demonstrate that our MWAM is highly effective and scales successfully to large-scale datasets, confirming its practical applicability.

---

### Meta-Review · Area_Chair_v6dg · 2025-12-26

**Summary:**

This paper addresses the critical problem of performance degradation in multimodal models when certain modalities are missing at inference time. The authors identify that it often stems from imbalanced learning, where dominant modalities hinder others during training. To mitigate this, they introduce the Frequency Ratio Metric (FRM) and the Multimodal Weight Allocation Module (MWAM) . The primary reason for acceptance is the novelty of the frequency-domain analysis used to diagnose and correct modality imbalance. This approach is not only intuitive but also results in a lightweight, and the empirical results across various tasks demonstrate consistent improvements.

**Reviewer Concerns:**

I think the rebuttal process successfully addressed the majority of the reviewers' technical reservations.

$\cdot$ For Lu1c: The initial concerns regarding the lack of rigorous theoretical justification and experiments on larger datasets were effectively addressed . The authors pointed to detailed theoretical proofs in the appendix regarding optimization dynamics and provided new results on the large-scale dataset .

$\cdot$ For bfT5: This reviewer questioned the generalizability of the method to non-visual modalities like audio or text. The authors clarified that the current scope is explicitly multimodal image understanding. I agree with the reviewer that this is a limitation.

$\cdot$ For K6MD: This reviewer initially provided a very low rating. Following an intensive four-round discussion, the authors clarified the benefits of DCT’s spatial layout for energy partitioning. K6MD eventually acknowledged the work's considerable potential and stated their intention was not to reject the manuscript but to help improve it.

I think the reviewers have reached a general consensus regarding the technical contribution and practical utility of the proposed module.

**Reviewer Scores:**

Based on the extensive rebuttal and the shift in the discussion:
Reviewer Lu1c, gN31, and bfT5: These reviewers initially gave 6 (marginally above acceptance).
Reviewer K6MD: This reviewer’s initial score of 2 was a significant outlier. However, by the end of the discussion, K6MD explicitly noted that they recognized the contribution and novelty of the frequency-domain perspective, I believe the score would have moved to a 4 or 6.

---

### Decision · Program_Chairs · 2026-01-26

Accept (Poster)